# Geomagnetic spikes on the core-mantle boundary

Christopher Davies[1,2] & Catherine Constable[2]

Extreme variations of Earth's magnetic field occurred in the Levant region around 1000 BC, when the field intensity rapidly rose and fell by a factor of 2. No coherent link currently exists between this intensity spike and the global field produced by the core geodynamo. Here we show that the Levantine spike must span $>60°$ longitude at Earth's surface if it originates from the core–mantle boundary (CMB). Several low intensity data are incompatible with this geometric bound, though age uncertainties suggest these data could have sampled the field before the spike emerged. Models that best satisfy energetic and geometric constraints produce CMB spikes 8–22° wide, peaking at $O(100)$ mT. We suggest that the Levantine spike reflects an intense CMB flux patch that grew in place before migrating northwest, contributing to growth of the dipole field. Estimates of Ohmic heating suggest that diffusive processes likely govern the ultimate decay of geomagnetic spikes.

[1] School of Earth & Environment, University of Leeds, Leeds LS2 9JT, UK. [2] Institute of Geophysics and Planetary Physics, Scripps Institution of Oceanography University of California at San Diego, 9500 Gilman Drive, La Jolla, California 92093–0225, USA. Correspondence and requests for materials should be addressed to C.D. (email: c.davies@leeds.ac.uk).

A key challenge to understanding the geodynamo process is to characterize and explain the most extreme spatial and temporal variations of Earth's magnetic field. Dipole dominance, high- and low-latitude intense flux patches and weak variations in the Pacific hemisphere[1,2] are robust characteristics of the field over the past few hundred years. Similar features have been reproduced in geodynamo simulations[3] and have been used as criteria for assessing whether simulations produce Earth-like behaviour[4–6]. Whether these kinds of spatial variations can reflect long-term behaviour of the geomagnetic field requires detailed observations preceding the historical period.

The highest geomagnetic field intensities on record have been recovered from archeomagnetic artefacts from the Levantine region dated at around 1000 BC. At this time the global field was unusually strong, with an increasing[7–9] axial dipole moment (ADM) of ∼95–100 ZAm². Yet the intensities recorded in Jordan[10] and Israel[11] around 980 BC correspond to local virtual ADMs (VADMs) of approximately 200 ZAm². The detection of high VADMs in Turkey to the North[12], in Georgia[13] to the East, and highs 150–300 years earlier in China to the North-East[14] lends support for a somewhat broader regional extent for this extreme feature, and an increasing number of high quality intensity data from distinct archaeological sites in Syria[15] provide additional temporal context for Middle Eastern intensity variations over the past 9,000 years. However, the morphology and spatial extent of the Levantine spike are presently unknown. Recent work suggests the existence of a second geomagnetic spike in North America that may be coeval with the Levantine spike[16]. Here we focus on the Levant region before discussing the issue of multiple spikes.

The Levantine geomagnetic spike is shown in Fig. 1, which presents six sets of spatially binned paleointensity data from sites with ages younger than 5000 BC and northern hemisphere locations from 15 to 60° E as downloaded from the online Geomagia.v3 database[17] (http://geomagia.gfz-potsdam.de/) on 4 December, 2015. We use the VADM representation for the data to minimize geographical effects due to axial dipole variation and show all data with age and intensity uncertainties as assigned (or not) by the original authors. The spike is clearly visible in the 10–40° N, 30–45° E geographic bin in Fig. 1e, but not elsewhere. Previous attempts to relate the rate of field changes inferred from the spike to fluid flow at the top of the core require velocities that are much faster than those corresponding to present secular variation and flow morphologies that are very different to those obtained from frozen flux inversions of geomagnetic data or from the present generation of geodynamo simulations[18,19].

The 2,138 archeomagnetic intensity data shown in Fig. 1 form a subset of a much larger globally distributed collection of observations (>80,000 over the interval 8000 BC to 1660 AD), that is dominated by directions and relative paleointensities from sediments and has been used to produce recent Holocene field models. A selection of maps of radial magnetic field, $B_r$, at the core–mantle boundary (CMB) are predicted from snapshots of the CALS10k.2 model[9] and shown in Fig. 2. From 2000 BC, flux at high northern latitudes moves equator-ward, slightly weakening the dipole. By 1500 BC a small flux patch has emerged under NE Africa/Saudi Arabia, which intensifies until 1000 BC before later moving north and west. This patch may be related to the Levantine spike although there is no obvious signature in the surface intensity (Fig. 4b), and the VADM

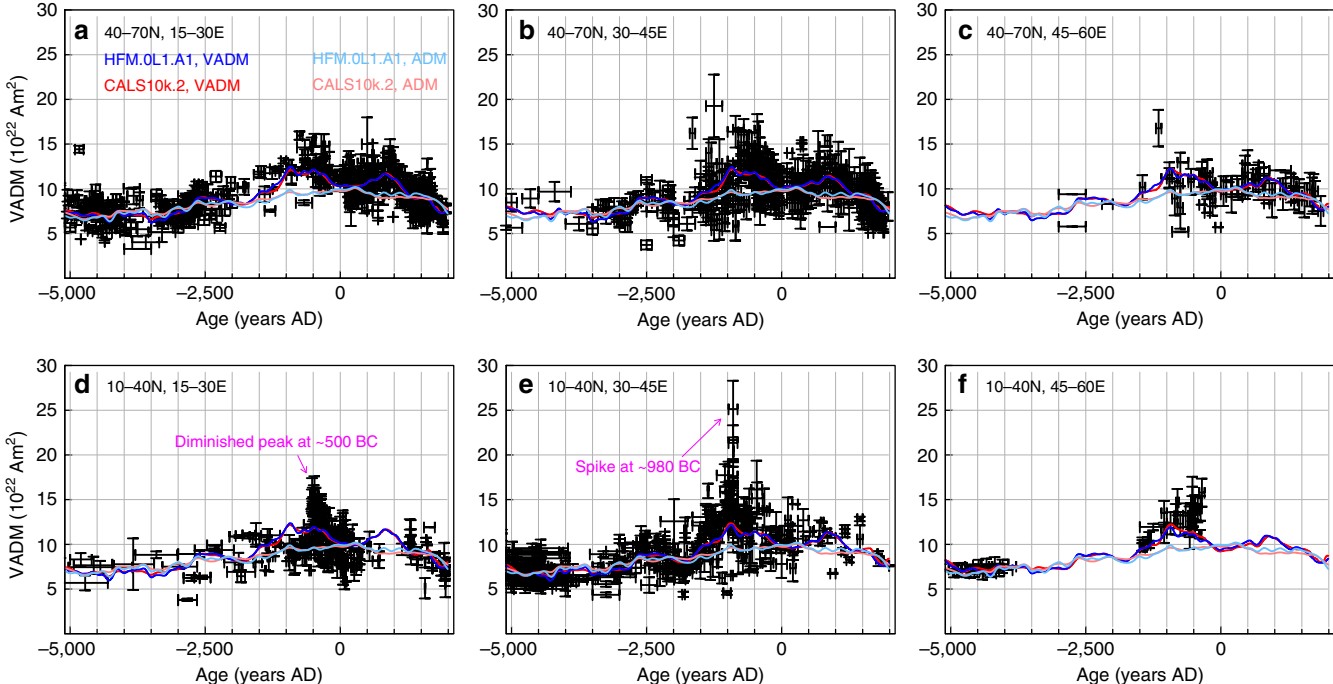

**Figure 1 | Temporal evolution of the Levantine spike.** Archeomagnetic and volcanic VADM data drawn from the Geomagia.v3 database[17] and plotted in geographic bins as noted in individual panels. Error bars are as assigned by the original authors, and where given are typically ±1 s.d. for radiocarbon ages, and ±1 s.d. about the mean VADM. However, in some publications the methodology is not reported and in some cases errors in both ages and intensity may be ±2 s.d. or 1 s.e. Please see the Geomagia.v3 database[17] for more detailed information on the errors. (**a**–**c**) present a longitudinal transect from 15 to 60° E, and data from 40 to 70° N in latitude, and **d**-**f** are the same longitude bins as directly above, but for latitudes 10–40° N. The spike is clearly visible in **e** at about 1000 BC, while other bins (except **c** which is more or less flat) exhibit lower peaks around 500 BC. Light coloured lines represent predictions of axial dipole moment (ADM) from recent Holocene field models[9], CALS10k.2 (red) and HFM.OL1.A1 (blue), while darker blue and red colours are VADM predictions for the average location of data in each bin. ADM increases to a peak value at about 300 AD, while VADM predictions are more regionally variable but still do not match the peak of the spike.

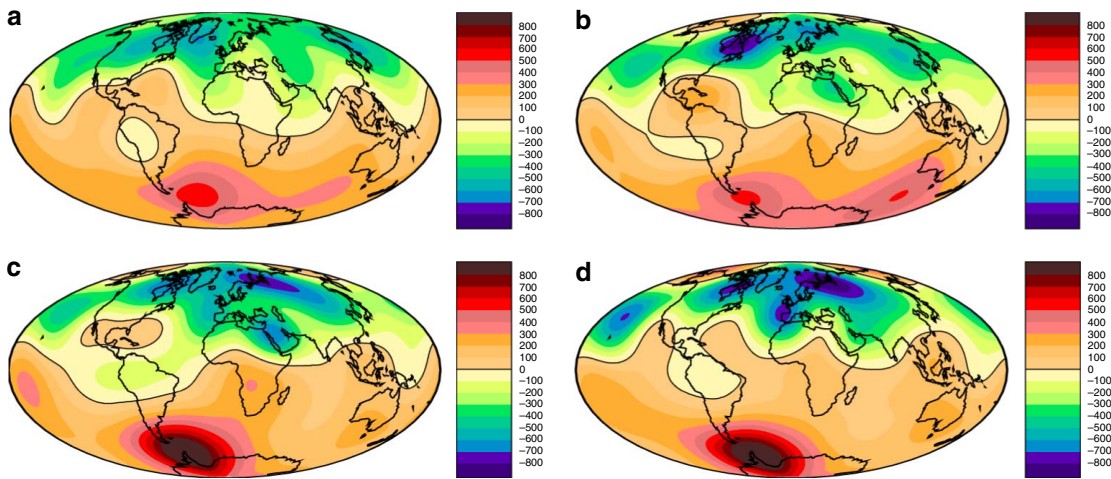

**Figure 2 | Global reconstructions of the geomagnetic field at times that span the Levantine spike event.** Maps of $B_r$ at the core–mantle boundary (CMB) predicted from model CALS10k.2 at (**a**) 2000 BC, (**b**) 1500 BC, (**c**) 1000 BC and (**d**) 750 BC. All scales for $B_r$ range from $-900$ to $900\,\mu$T. Note the incipient flux patch under Saudi Arabia at 1500 BC in **b** that grows into a relatively strong flux patch centred on 25° N in **c**. By 750 BC it appears to have moved west and north to merge with a patch lying to the south of the UK. There is no spike signature in the associated surface intensity (Fig. 4b), because of low resolution in the CALS10k.2 model.

predicted by CALS10k.2 is much weaker than the observed values (Fig. 1e) around 1000 BC.

In this study we first map the spatial extent of the Levantine spike, showing that the region of anomalously high intensities, where the field strength rose and fell by a factor of 2, is confined to only 20° longitude. We then create a model of the spike by assuming, as in previous studies[10,11,18–20], that it is a product of the geodynamo process that generates Earth's internal magnetic field through motions of the liquid iron core. In this model the synthetic spike is entirely determined by its amplitude, width and location, all specified at the CMB. We find that even the thinnest CMB spikes ($<1°$ wide) produce a surface feature that spans at least 60° in longitude. Models that trade off matching the surface spike intensity, minimizing $L^1$ misfit to the available data, and satisfying core energy constraints produce CMB spikes 8–22° wide, peaking at O(100) mT. Such extreme behaviour is not observed in global models of the recent magnetic field and does not appear to have been identified in the current generation of geodynamo simulations.

## Results

**Spatial map of the levantine geomagnetic spike.** To create an approximate spatial map of the Levantine spike seen at Earth's surface we consider the Jordanian and Israeli data to be coeval and centred at 1000 BC. Although there is evidence for two distinct spikes separated by 1–2 centuries in the observations[11] we do not make this distinction here, instead focussing on a single potential deep-seated cause at the CMB. We assigned an uncertainty to the spike age of $\pm100$ years, which is similar to the age uncertainty on the Jordanian data and to the average age uncertainty of the paleointensity data used to construct CALS10k.2. We then selected all absolute intensity data from the Geomagia.v3 database[17] with assigned ages in the range $1000\pm150$ BC, yielding 143 data in 23 locations. Data without any prescribed age uncertainty were assigned the average value from the CALS10k.2 paleointensity data, namely $\sigma_{age}=110$ years (Fig. 3a). A given datum is considered to be coeval with the spike if their ages overlap within the uncertainties.

Further data selection is hampered by the significant complexities associated with paleointensity error estimation. Differences in host material, conditions at the time of remanence acquisition, laboratory protocols, age controls and documentation mean that

it is very hard to define paleointensity errors that can be compared across different samples[21,22]. Improvements to paleointensity determination inevitably mean that older samples were not subjected to the same strict selection criteria used recently on the Levantine artefacts. In view of these difficulties we adopt the reasonable strategy of retaining all of the data, thus achieving critical mass and avoiding potential bias associated with specific protocols at the expense of requiring robust $L^1$ measures for data misfit. An alternative approach would be to reject data where the paleointensity uncertainty is anomalously high. We compared the age and intensity uncertainties for all archeomagnetic paleointensity data used to construct CALS10k.2 (Fig. 3). The overall intensity uncertainty was assumed to arise from three factors (Methods): intensity uncertainty as a product of the laboratory measurements, $\sigma_{lab}$; age uncertainty based on dating the sample, $\sigma_{age}$; and age difference, $\Delta$, between the sample age and the time of the spike, taken to be 1000 BC. The total uncertainty $\sigma_t^2 = \sigma_{lab}^2 + (\partial F/\partial t)^2\left[\sigma_{age}^2 + \Delta^2\right]$, where $\partial F/\partial t$ is the rate of intensity change. Values of $\sigma_{lab}<5\,\mu$T were considered to be unrealistically low[9] and so these data were assigned an uncertainty of $\sigma_{lab}=5\,\mu$T. Representative values of $\sigma_t$ calculated using $\partial F/\partial t=0.15\,\mu$T yr$^{-1}$, similar to the historical geomagnetic field[1], provide evidence for the influence of age uncertainties (Fig. 3b, Supplementary Table 1). Nevertheless, the 143 global data that constrain the spatial structure of the Levantine spike are clearly typical of the overall CALS10k.2 data set, reinforcing the view that there is no justification for further data rejection at this stage.

To visualize the spike, we select at each of the 23 locations the datum with peak intensity (Fig. 4b–d, Supplementary Table 1). Some low values occur in the Levantine region at times nominally ranging from 1150 BC to 1025 BC (open symbols in Fig. 4d). Age uncertainties mandate their inclusion here, highlighting the issue that constraining absolute ages across the region remains difficult despite the high quality stratigraphic constraints possible within some individual sites[11]. Nevertheless, the spike is remarkably localized: normal intensities in Bulgaria[23], India and Ukraine suggest a longitudinal extent of only 20°. There are no high-intensity features comparable to the spike in satellite[2], historical[1] or Holocene[7,8] geomagnetic field models (compare Fig. 4a–c).

**Model of the levantine spike.** We assume that the geomagnetic spike is the surface expression of short wavelength structure in the radial magnetic field, $B_r$, at the CMB (Methods). We do not consider how the spike might be produced, only the consequences of its existence. Holocene field models like CALS10k.2 (Figs 2c and 4b) are too smooth to show the spike in the field intensity at Earth's surface[7,8] and so we create a synthetic spike field, $B_r^{spike}$, that is combined with the field $B_r^o$ predicted from a representative model (Fig. 4b). A representation of the spike is required in both physical space using spherical polar coordinates ($r$, $\theta$, $\phi$), and in wavenumber space using spherical harmonics of degree $l$ and order $m$. This motivates the circular Fisher–Von Mises probability density function as a suitable choice for the spatial form of $B_r^{spike}$. A spike centred at ($\theta^c$, $\phi^c$) in spherical coordinates is then defined entirely by the amplitude $A$ and s.d. $\sigma$ of the distribution. Assuming an insulating mantle, the total radial field is simply $B_r = B_r^o + B_r^{spike}$.

Values of $A$, $\sigma$, $\theta^c$ and $\phi^c$ are varied simultaneously to match the observed spike intensity and width. Any single peak in intensity can always be produced by varying $A$ and $\sigma$. Spike width is defined with respect to the peak intensity $F_{max}(r) = F(r, \theta^c, \phi^c)$. The latitudinal extent of the spike is hard to estimate, partly because there is little coeval data north and south of Jordan and Israel and partly because of the latitudinal increase in dipole field strength. We therefore define the regional spike width $\delta_x(r)$ as the longitudinal range over which $F > F_{max}/x$, where $x$ is a parameter that is $>1$. At 1000 BC the CALS10k.2 and PFM9K field models[7,8] give $F(a) \approx 60–65\,\mu T$ at latitude $\lambda = 30°$ north, where $a$ is Earth's radius; together with the value of $F_{max}(a)$ from Fig. 4d, we obtain $x \approx 1.6–2.0$. Individual data (Fig. 4b) yield a similar result and so we focus on $x = 2$, although using $x = 1.6$ makes little difference.

With $B_r^o = 0$, decreasing $\sigma$ towards zero reduces the width of the surface spike towards a minimum of $\delta_2(a) \approx 60°$ (Supplementary Figs 1–3), much larger than the observed value of $\delta_2(a) \approx 20°$. This behaviour can be understood by considering the limit of a delta function spike ($\sigma \rightarrow 0$) at the CMB, which has equal power at all wavenumbers. The field decays with increasing $r$ as $(a/r)^{l+2}$ (Methods) and so the small-scale structure that is added to the CMB spike by decreasing $\sigma$ is geometrically attenuated; the field exhibits no spike-like feature at Earth's surface (Fig. 5). A suite of models with $B_r^o = 0$ predict that the dominant surface expression of the spike is a change in the $l = 1$ dipole component of the field for all $0.5° \leq \sigma \leq 40°$ and $200 \leq A \leq 800\,mT$, although in relative terms it does not make a large immediate change to an existing dipole moment. With $B_r^o$ defined by the CALS10k.2 field model at 1000 BC the spike is much wider, even for small values of $\sigma$, owing to lateral intensity variations in the field model (compare Supplementary Fig. 3 and Fig. 4d).

The predicted minimum spike width cannot be further reduced by adding time dependence or dynamics associated with the dynamo to the model. The problem is purely geometrical: the source is very far from the observation point. Allowing for finite mantle conductivity will smooth and delay core signals[24], but does not move the source significantly closer to the surface because the mantle is thought to be weakly conductive everywhere except in a thin layer above the CMB[25]. The case of multiple closely spaced CMB spikes separated by a distance $\Delta$ at the CMB has also been investigated. When $\Delta < 5°$ the corresponding surface feature appears as one large spike (Supplementary Figs 1 and 3). With $\Delta \approx 5°$ the surface feature has a flat top due to smearing of adjacent spikes. At large separations (for example, $\Delta = 15°$, Supplementary Fig. 2). the individual spikes are visible as a broad high-intensity region at the surface. In principle, a CMB magnetic flux distribution consisting of concentric rings of alternately signed flux would create a

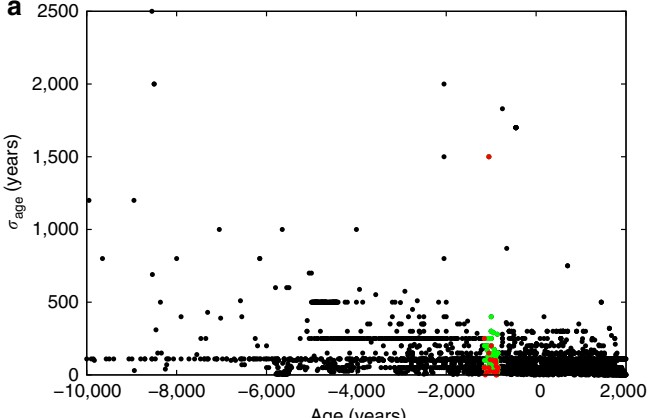

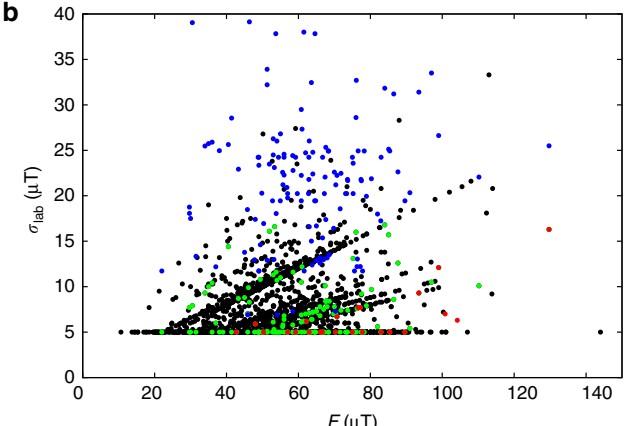

**Figure 3 | Estimated uncertainties on paleointensity data used to constrain the Levantine geomagnetic spike.** (**a**) Black circles give age uncertainty estimates, $\sigma_{age}$, for 4,128 paleointensity estimates spanning the interval 0–10 ka used in the construction of the CALS10k.2 Holocene field model[9] as described in the text. Green circles highlight the 143 data lying in the interval 1150–850 BC. Red circles mark estimates associated with the peak value at each of the 23 sites used in Fig. 4. (**b**) Black, green and red symbols as in the upper panel but now showing $\sigma_{lab}$. Blue circles show $\sigma_t$, which incorporates uncertainty effects arising from dating ($\sigma_{age}$) and age bias $\Delta$ arising from the discrepancy in the assigned age relative to the date of the Levantine spike, taken here to be 1000 BC (Methods). The total uncertainty is $\sigma_t^2 = \sigma_{lab}^2 + (\partial F/\partial t)^2 [\sigma_{age}^2 + \Delta^2]$, with the rate of intensity change $\partial F/\partial t$ taken to be $0.15\,\mu T\,yr^{-1}$.

narrow high-intensity field structure at the Earth's surface, however, such a configuration is not thought to be likely for a dynamo-generated field and has not been investigated here. The thinnest surface feature is obtained with only one spike at the CMB (Supplementary Figs 3 and 4). This, together with the available data distribution (Fig. 4), strongly suggests that the spike cannot be centred under the Near East.

To constrain spike morphology and position we generated 750 models with $B_r^o$ defined by the CALS10k.2 field model at 1000 BC and a range of amplitudes, widths and locations ($A$, $\sigma$, $\theta^c$ and $\phi^c$). Normalized model misfit for the $n = 143$ data ($F_d^i$, $i = 1 \ldots n$) was initially evaluated using both $L^1$ and $L^2$ measures of difference from the model predictions, $F_m^i$. We define $L^1 = n^{-1} \sum_i^n \left| (F_d^i - F_m^i)/\sigma_t^i \right|$ and $L^2 = \sqrt{n^{-1} \sum_i^n \left| \frac{F_d^i - F_m^i}{\sigma_t^i} \right|^2}$ for which we should expect values of $L^1 = 1.128$ and $L^2 \sim 1$, respectively[26] if we have accurate estimates of $\sigma_t^i$. The $\sigma_t^i$ are calculated as described above (Fig. 3) both with and without the

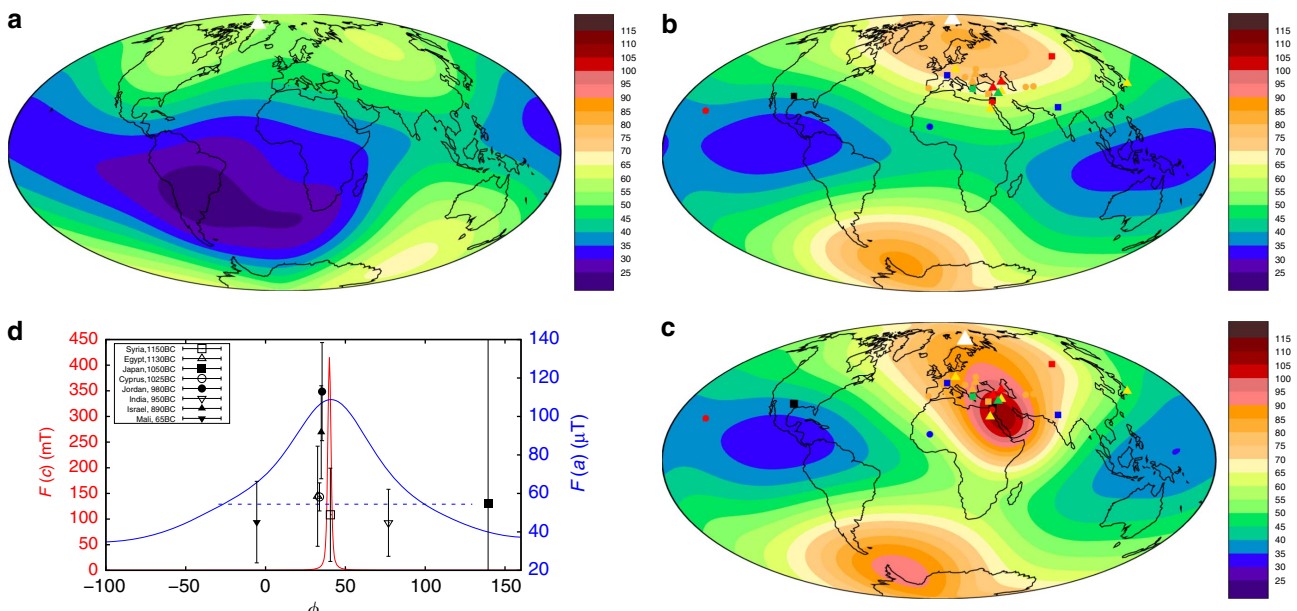

**Figure 4 | The Levantine geomagnetic spike.** Contours of field intensity, $F$ ($\mu$T), at Earth's surface ($r = a$) from the CHAOS-4 model[2] at 2010 (**a**) the CALS10k.2 global field model at 1000 BC (**b**) and CALS10k.2 at 1000 BC plus a superposed best-fitting spike at 20° N, 40° E with amplitude $A = 400$ mT and s.d. of $\sigma = 1°$ at the CMB (**c**). Symbols show paleointensities for samples dated at 1150–1050 BC (triangles), 1049–950 BC (squares) and 949–850 BC (circles). Symbol colours are blue (40–50 $\mu$T), green (51–60 $\mu$T), yellow (61–70 $\mu$T), orange (71–90 $\mu$T), red (91–115 $\mu$T) and black (>115 $\mu$T). White triangles in **a**–**c** mark the north pole of the dipole field. (**d**) Longitudinal cross-section through the spike in **c**, at Earth's surface (blue, right ordinate) and the CMB ($r = c$, red, left ordinate). The horizontal dashed line marks the width at half maximum $\delta_2(a)$. Available data within 20° ± 15° N are shown corrected to 20° N using the formula for an axial dipole field, $F \propto \left(1 + 3\cos^2\theta\right)^{0.5}$, where $\theta$ is colatitude. Error bars correspond to the uncertainties in Fig. 3b. Open and closed symbols cannot be simultaneously matched by the model.

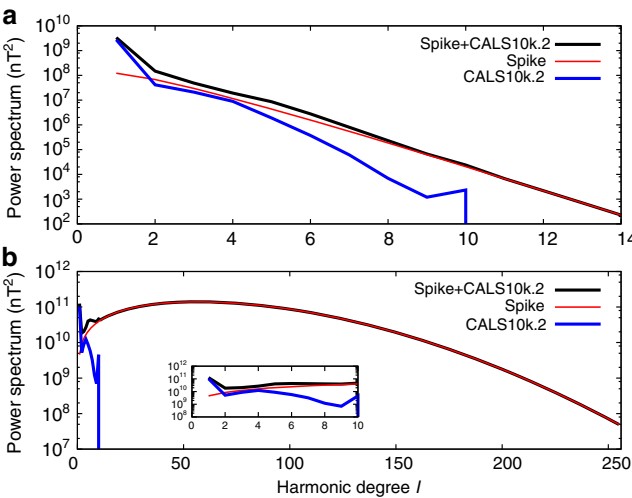

**Figure 5 | Power spectrum of a geomagnetic spike.** Spatial power spectrum $R_l$ for CALS10k.2 at 1000 BC, a representative spike, and spike plus CALS10k.2 as a function of spherical harmonic degree $l$ at Earth's surface (**a**) and the core–mantle boundary (CMB) (**b**). The parameters for the spike model are the same as those in Fig. 4. Note the different scales for the abscissa in **a,b** and the attenuation of power at high $l$ as the radius increases from the CMB to the surface. The inset in **b** shows a blow-up of the range $l = 0$–10.

age bias term $\Delta$ and for various values of $\partial F/\partial t$. Since models with minimum $L^1$ (and $L^2$) are too smooth to fit the high-intensity Near East data we also require that a suitable model fits the observed intensity at the location of the spike, initially taken to be Israel.

The presence of outliers precluded obtaining reasonable fits using the $L^2$ quadratic misfit measure. We therefore focus on $L^1$. With $\partial F/\partial t = 0.15\,\mu$T yr$^{-1}$ (Fig. 3) the best fits to the spike correspond to $L^1 \sim 2$. This is higher than expected based on our uncertainty estimates. It is possible that the average lab uncertainties in paleointensity measurements of 5–10% are too low owing to the issues described above[21]. Alternatively, setting $\partial F/\partial t = 0.15\,\mu$T yr$^{-1}$ could be too conservative during a time of rapid dipole growth and occasional extreme regional field variations and this is supported by the rates of change of $\geq 1\,\mu$T yr$^{-1}$ inferred from data in the Levant region around 1000 BC (refs 18,27). A small increase of $\partial F/\partial t$ to 0.2 $\mu$T yr$^{-1}$ gives values of $L^1$ within the acceptable range (Fig. 6), which does not seem unreasonable. Larger values of $\partial F/\partial t$ shift the data in Fig. 6 to lower $L^1$ since this gives more weight to the age uncertainties, which already represent the main contribution to $\sigma_t$ for $\partial F/\partial t = 0.15\,\mu$T yr$^{-1}$.

Adding a synthetic spike to the CALS10k.2 1000 BC field improves the fit to the high-intensity Levantine data while retaining a satisfactory $L^1$ misfit to the overall data set (Fig. 6). There is a relatively poor fit to some data (for example, Hawaii, Mali, Switzerland, China) whether or not the spike is present (Fig. 4b,c), reflecting the localized nature of the spike field. However, even the best-fitting spikes (Fig. 4c) are unable to match both the low (for example, Syrian, Egyptian and Cypriot) and high-intensity data in the Levantine region. This could be because the spike feature changed so fast[18] that the approach of selecting all data in a 300 year window bracketing the main spike event is incorrect, though very rapid changes are not compatible with present core flows[18] based on the frozen flux approximation. A potentially more satisfactory explanation is that the discrepancy arises due to inaccuracies in the dating and age bias (note that Syrian, Egyptian and Cypriot data are all nominally at older ages

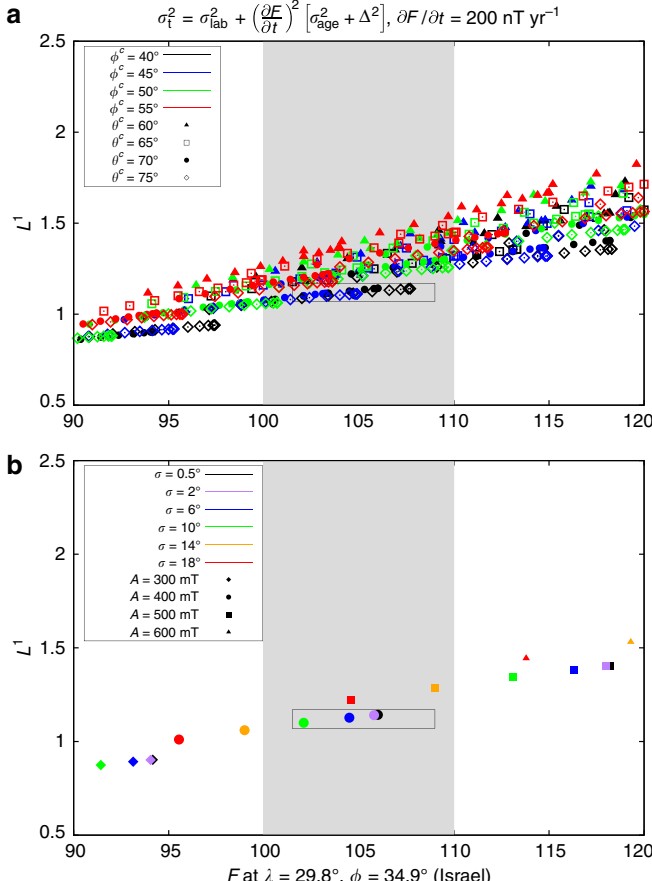

**a** $\sigma_t^2 = \sigma_{lab}^2 + \left(\frac{\partial F}{\partial t}\right)^2 \left[\sigma_{age}^2 + \Delta^2\right], \partial F/\partial t = 200 \text{ nT yr}^{-1}$

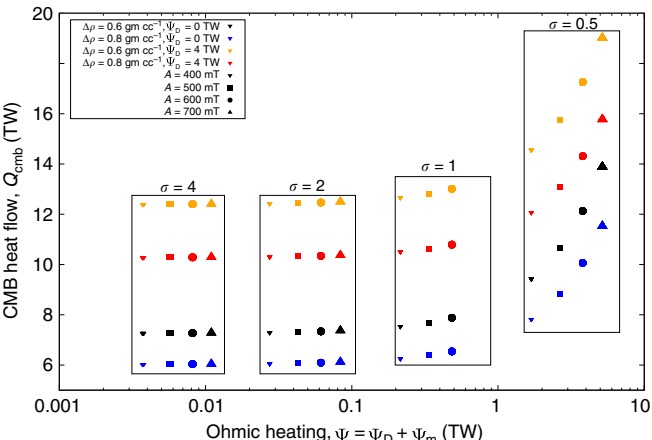

**Figure 7 | Impact of a geomagnetic spike on the core energy budget.** Thinner (lower $\sigma$) and stronger (higher $A$) spikes produce more Ohmic heating $\psi_m$ that, in steady state, require a faster cooling rate and larger CMB heat flow for their maintenance.

**Figure 6 | Misfit to the Levantine spike data.** Two representations of the same 750 models with different parameter dependencies highlighted. The grey shaded region marks the spike intensity in Israel[10] with generous error bars applied. Best-fitting models, outlined by the rectangles, provide the best trade-off between matching the spike intensity and minimizing $L^1$ misfit. (**a**) Highlights dependence on location: $\phi^c$ is indicated by colours while $\theta^c$ is shown by shape. (**b**) CMB spike location is fixed at $\theta^c = 70°$ and $\phi^c = 40°$ to illustrate dependence on $A$ and $\sigma$. Models with thin spikes (low $\sigma$) located south east of Jordan are preferred.

than the Jordanian and Israeli data, while data in Mali and India are nominally younger). What is clear is that the high and low Levantine intensity data cannot be simultaneously matched by changing the spike geometry.

Best-fitting model spikes are obtained when the spike is centred south East of Jordan around $(\lambda^c, \phi^c) = 20°$ N, $40°$ E (Figs 4 and 6b), close to the position of a relatively strong CMB flux patch in the CALS10k.2 field (Fig. 2c). This location is favoured because it reduces the field intensity in south East Europe, which tends to be slightly over-predicted. Supplementary Fig. 5 shows that the best trade-off between minimizing weighted $L^1$ misfit and fitting the spike intensity in Israel is achieved for the same parameter combinations regardless of the weighting applied in the misfit calculation. We, therefore, consider that the constraints on spike location and geometry shown in Fig. 6 are plausible given the present availability and quality of data.

At the CMB the best-fitting model spikes are strong and thin, with $400 \text{ mT} \leq A \leq 600 \text{ mT}$ and $0 < \sigma < 18°$ The corresponding CMB field intensities range from O(1) mT for very wide spikes ($\sigma = 18°$), comparable to the peak historical field[1], to 1 T for thin spikes ($\sigma = 0.5°$). While the latter value seems absurdly high it cannot be ruled out using surface magnetic field observations alone as long as the CMB field is so localized that it is obscured

from detection (that is, there is significant power at harmonic degrees beyond $l \approx 20$ as in Fig. 5). The present root mean square (RMS) CMB field strength, $B_r^{RMS}$, including the small scales, can be inferred using numerical geodynamo simulations[28] and studies of Earth's nutations[29,30] and length-of-day variations[31], and may be as large as 1 mT. Assuming a similar value around 1000 BC eliminates only the very thin spikes: all models with $\sigma \geq 4°$ and $A \leq 600 \text{ mT}$ have $B_r^{RMS} < 1.7 \text{ mT}$.

Significant regional variations in field direction are predicted by our model (Fig. 8), which could in principle provide additional constraints on its validity. However, the archeomagnetic slag samples used to identify the original intensity spike are unsuitable for accompanying directional measurements, while only two results were available in the Geomagia database between 1150 and 850 BC in the geographic bin containing the spike (Fig. 1e). Recent work on samples from fired ovens from Tel Megiddo, Israel[11] have recovered steep inclination values of up to 75° at around the time of the spike, which are unusual at that latitude, while Turkish data with high VADMs are accompanied by slightly lower inclinations than in Israel suggesting they might lie slightly further from the centre of the spike. The predicted model directions are in line with these limited observations.

A final independent constraint on the spike is obtained by estimating its contribution to the core energy budget. In a steady dynamo, magnetic energy created through work done by fluid motions on the field is balanced by energy lost via Ohmic heating, denoted $\Psi$. Together with estimates of core physical properties, $\Psi$ constrains the core cooling rate and provides an estimate of the CMB heat flow, $Q_{cmb}$, which can be compared to independent estimates[32] of $Q_{cmb} = 5$–$15$ TW. We use a standard model for the energy budget[33] and two sets of core material properties[34] that were calculated for two different values of the inner core boundary density jump[35], $\Delta\rho = 0.6$ and $0.8 \text{ gm cc}^{-1}$ (ref. 36) (Methods). We estimate the $\Psi$ due to the spike using a solution for the minimum Ohmic heating associated with the observable magnetic field[37], denoted $\Psi_m$. This estimate omits $\Psi_D$, the part of $\Psi$ due to the main dynamo process, and so provides a conservative estimate of $Q_{cmb}$. Estimates of $\Psi_D$ range from 0.5 to 10 TW (ref. 38), when accounting for recent results giving an increased core electrical conductivity[39]. We take $\Psi_D = 4$ TW as an example and assume $\Psi_m$ and $\Psi_D$ can be added to obtain a simple estimate of $\Psi$ (Methods).

With $\Psi_D = 0$ none of our spikes require a CMB heat flow exceeding 15 TW, while with $\Psi_D = 4$ TW only the strongest

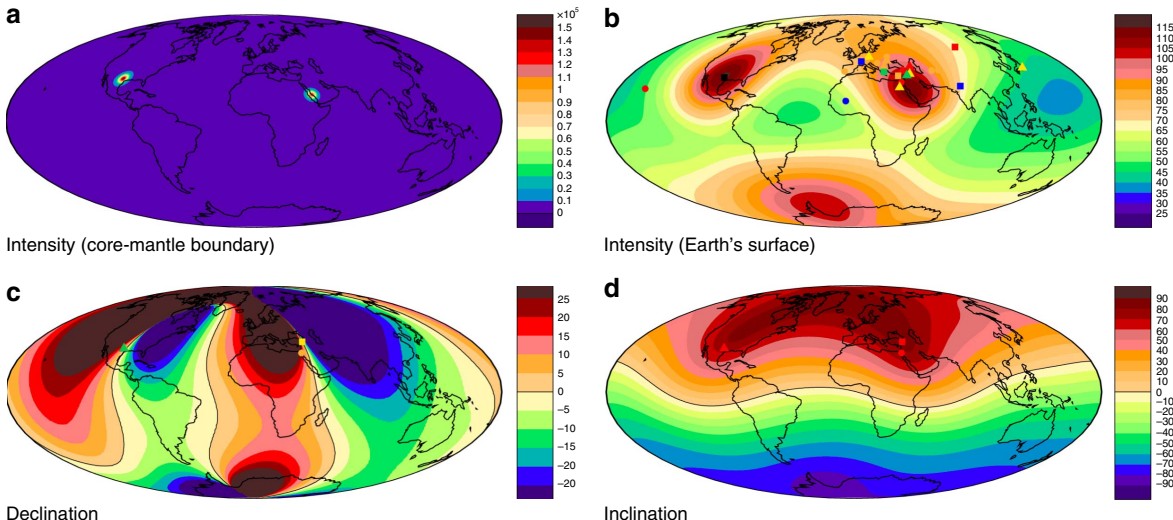

**Figure 8 | Intensity and directional predictions resulting from geomagnetic spikes under Texas and the Levant.** Intensity $F$ in µT at the core–mantle boundary (**a**), and Earth's surface (**b**) for a model with one geomagnetic spike under Texas ($A = 600$ mT, $\sigma = 1°$) and another spike under the Levant ($A = 400$ mT, $\sigma = 1°$). Maps of declination (**c**) and inclination (**d**) in degrees at Earth's surface are also shown. Symbols in **b** are the same as in Fig. 4. Symbols in **c,d** show $I = 50$–$53°$ and $D \approx -10°$ for Halls Cave (Fig. 9 in ref. 16), $I = 65$–$75°$ and $D \approx -5$–$20°$ in Israel (Fig. 6 of ref. 11), and $I = 65°$ and $D \approx 0°$ in Turkey (Fig. 7 in ref. 12).

($A \geq 600$ mT) and thinnest ($\sigma = 0.5°$) spikes require $Q_{cmb} > 15$ TW (Fig. 7). The presence of the spike is only apparent in the gross energetics for $\sigma < 2°$, where the CMB heat flow required to maintain the spike alone is the same order of magnitude as that required to maintain the whole dynamo. We view this as unlikely and so favour spikes with $\sigma > 2°$ at the CMB. Relaxing the steady state assumption does not change the results (Methods).

Applying the constraints provided by paleomagnetic data, estimates of CMB field strength and the core energy budget to the best-fitting synthetic spikes (marked by rectangles in Fig. 6), we infer the model parameter values $A \approx 400$ mT and $\sigma \approx 4$–$10°$. These values correspond to a CMB spike width of $\delta_2(c) \approx 8$–$22°$ and a surface width of $\delta_2(a) \approx 60°$.

It is interesting to consider how the above results are affected by the presence of multiple CMB spikes. We consider the specific case of a recent sediment record from Texas[16], which is apparently coeval with the Levantine spike and also shows rapid changes to very high field strength. Assigning different amplitudes and widths to a pair of spikes allows us to obtain models with slightly lower $L^1$ misfits and a $\sim 10\%$ increase in ADM compared to models with a single spike under the Levant. For example, the model in Fig. 8 has $L^1 = 0.46$, a dipole moment of $116$ ZAm$^2$ and ADM of $115$ ZAm$^2$ compared to $L^1 \approx 1.18$, dipole moment of $105$ ZAm$^2$ and ADM of $102$ ZAm$^2$ for the single-spike model in Fig. 4c. This change in ADM is within the errors of the data[8,9], while the associated directional deviations are compatible with the limited directional data accompanying recent studies[11,12,16] (Fig. 8). As in the single-spike case the Ohmic heating can be kept to an acceptable level as long as neither spike is too thin.

## Discussion

Our model shows that the instantaneous surface expression of high intensities must extend over a broad geographical area, estimated to be $\approx 60°$ longitude. Some of the data in Fig. 4 are inconsistent with this result, but the issue cannot be rectified by appealing to unresolved small scales in the core field. Ideally we would identify low-quality intensity estimates and those with poor age control and remove them from the analysis. The estimates presented in Supplementary Table 1 indicate that total

uncertainties exceeding 45% of the measurement (with $\partial F/\partial t = 0.2$ µT yr$^{-1}$) pertain to data from Japan, Mali, Czech Republic, India, Greece, Syria and Egypt. It is tempting to remove these data, since this would all but resolve the inconsistencies in the Near East shown in Fig. 4. However, as described above, quantification of uncertainty remains a challenge even with gold standard paleointensity protocols. Our uncertainty assessment has its own limitations, including reliance on the consistency of individual laboratory and dating errors and the poorly known rate of intensity change in the Levant around 1000 BC. The simplest explanation, suggested by the sample dates and age uncertainties, is that the low intensity samples found in Syria, Egypt and Cyprus sampled the field before the emergence of the spike.

On the basis of synthetic spike models that provide the best fit to the available data and assuming that low intensity samples in the region did not sample the spike field we suggest that the Levantine geomagnetic spike reflects an intense and localized near-equatorial flux patch on the CMB, perhaps reminiscent of features seen in the modern field[40]. The available data east and west of the Levant both suggest a VADM peak of $\sim 150$ ZAm$^2$ around 500 BC (Fig. 1d,f), which argues against pure east–west drift of this patch, while flux patches produced by rotationally constrained dynamos do not usually cross the equator[41]. This suggests that the patch either emerged from within the core and subsequently decayed in the same place, or grew in the equatorial region and migrated northwards (and westward).

Equatorial growth and subsequent northward migration of the spike is an appealing scenario because this would contribute to the $\approx 15$–$20\%$ increase in dipole field strength[42] seen in Holocene field models at that time[7,8]. Changes in the present dipole field strength are predominantly due to meridional advection by a large-scale anti-cyclonic gyre[43]. We envisage that a similar circulation could have transported the Levantine flux patch westward in the equatorial region before advecting it northward around American longitudes. The equatorial flux patch under Saudi Arabia in the CALS10k.2 model appears to follow a similar evolution (Fig. 2), suggesting that this patch may be a low-resolution image of the Levantine spike, although the exact evolution of this feature remains uncertain owing to the available

temporal resolution of the CALS10k.2 field model[44,45]. In this scenario the tantalising evidence for an intensity spike recently identified in Texas[16] (interpreted as a second intense flux spot on the CMB) would also enhance the dipole, since normal flux under North America is located near the northward leg of the gyre and so would also be swept towards the pole[46].

A complication to this simple scenario is the possible occurrence of another Levantine spike[11] that emerged after the main peak in 1000 BC. Current evidence for this second later spike is not clear-cut: an initial determination of a second spike at 890 BC (ref. 10) was rejected based on stronger selection criteria[11], with new data suggesting a second spike around 800 BC. One plausible explanation is that these surface features reflect the propagation of multiple intense flux patches on the CMB. Alternatively, the two intensity highs might sample the edges of a single-flux patch.

Constraints on the spatial structure of geomagnetic spikes also provide additional insights into their temporal evolution. Theoretical calculations of rapid intensity changes require knowledge of the CMB magnetic field at scales that cannot be resolved by current observations[18]. Our results show that the power spectrum of the small-scale field associated with a spike (Fig. 5) cannot be obtained by a simple extrapolation of the observable field, which predicts decreasing power with increasing harmonic degree. It may also be possible to incorporate the energetic requirements for sustaining a given spike as constraints in core flow inversions. Furthermore, estimates of the Ohmic heating associated with geomagnetic spikes show that diffusion is likely to be important for the dynamics. Indeed, diffusion may ultimately be responsible for the decay of the spike as this process would be much faster than diffusive decay of a dipole field[39] because the spike is so thin. We speculate that radial diffusion, which has been hitherto neglected in evolutionary models of the spike, is crucial to reconciling the rapid observed temporal variations (Fig. 1) with flow at the top of the core. Since radial diffusion is not directly constrained by geomagnetic observations one possibility is to incorporate a parameterization of the effect in core flow models using results from numerical geodynamo simulations[47].

The present constraints on extreme geomagnetic field variations can guide the locations of future paleomagnetic acquisitions with a view to improving spatial coverage of the Levantine spike and to identifying new extreme geomagnetic events. Our results predict that the Levantine spike may have actually reached peak values south east of Israel, in which case paleointensity determinations from north-eastern Africa and Saudi Arabia would provide important constraints on the spike geometry. More directional data around 1000 BC, particularly east and west of the predicted spike location, are also essential to better quantify field behaviour in and around the Levantine region. The role of diffusion in controlling decay of geomagnetic spikes can, in principle, be tested in geodynamo simulations, although it remains to be seen whether the current generation of models produce the extremely spatially localized intensity variations that characterize the Levantine spike. Simulations can also be used to examine the origin of geomagnetic spikes. Potential mechanisms could include the expulsion of very strong toroidal flux concentrations[48] (though this might produce a pair of equatorially symmetric spikes, which is not presently observed), or the concentration of magnetic flux near the CMB by a convergent downwelling flow[49].

## Methods

**Treatment of paleointensity uncertainties.** We assume that three effects contribute uncertainty when using paleointensity data to constrain the Levantine geomagnetic spike: intensity uncertainty as a product of the laboratory measurements, $\sigma_{lab}$; age uncertainty based on dating the sample, $\sigma_{age}$; and age difference, $\Delta$, between the sample and the time of the spike (here taken as 1000 BC). The impact of both the latter contributions depend on the rate of change of the field and so the mean square error $\sigma_t^2$ can be written as $\sigma_t^2 = \sigma_{t1}^2 + \sigma_{t2}^2 = \sigma_{lab}^2 + (\partial F/\partial t)^2 \left[\sigma_{age}^2 + \Delta^2\right]$, where $\sigma_{t2}^2 = (\partial F/\partial t)^2 \Delta^2$. For an individual intensity datum the contributions to $\sigma_t^2$ were summed in quadrature. $\sigma_{lab}$ was initially set to the uncertainty assigned by the original authors where available. Assigned uncertainties of $<5\,\mu T$ were considered to be unrealistically low[9], and so these data were assigned an uncertainty of $5\,\mu T$. Data without intensity uncertainty were also assigned a value of $\sigma_{lab} = 5\,\mu T$. $\sigma_{age}$ was taken from the original publications where this information was provided, otherwise it was set to the average value for the entire CALS10k.2 paleointensity data set, $\sigma_{age} = 110$ years. The bias term $\sigma_{t2}^2 \sigma_t^2$ arises because the estimated age of some samples is quite different from the assumed age of the spike; these samples are still included since the age uncertainties are such that they could be coeval with the spike. We estimate $\Delta$ as the age difference between 1000 BC and the nominal age of the sample, and use this to evaluate bias expected because of secular variation arising from the age mis-match.

The rate of intensity change, $\partial F/\partial t$, in the Levant around 1000 BC is poorly constrained. Values as high as $4$–$5\,\mu T\,yr^{-1}$ have been postulated from the data[10], while theoretical bounds[18] based on the frozen flux approximation suggest $0.6$–$1.2\,\mu T\,yr^{-1}$. We consider three values of $\partial F/\partial t = 0.1$, $0.15$ and $0.20\,\mu T\,yr^{-1}$. The middle value is close to the maximum value found for the gufm1 historical field model[1] spanning the period 1590–1990 AD.

**Spike model.** We make the standard assumption that the mantle is an insulator, which means that the toroidal field is zero on the CMB (radius $r = c$). For $r > c$ the magnetic field $\mathbf{B}$ can be written as the gradient of a scalar $\psi$,

$$\mathbf{B} = -\nabla\psi, \tag{1}$$

and the constraint

$$\nabla \cdot \mathbf{B} = 0, \tag{2}$$

implies $\nabla^2\psi = 0$, which is Laplace's equation. In this case knowledge of the radial component $B_r$ is enough to determine $\psi$ everywhere[50]. The solution for $\psi$ is standard[37] and can be used to obtain the three components of $\mathbf{B}$. In spherical polar coordinates $(r, \theta, \phi)$ the radial component $B_r$ is given by

$$B_r(r, \theta, \phi) = a \sum_{l=1}^{\infty} \sum_{m=0}^{\infty} \left[c_l^m Y_l^m(\theta, \phi)\right] \left(\frac{a}{r}\right)^{l+2}. \tag{3}$$

Here $Y_l^m$ are spherical harmonics of degree $l$ and order $m$ and $a$ is the radius of Earth's surface. In practice the infinite series is truncated at $l = $ Lmax. The complex coefficients $c_l^m$ used in our code are related to the familiar Gauss coefficients $g_l^m$ and $h_l^m$ by $g_l^m = 2l\mathrm{Re}(c_l^m)$ and $h_l^m = -2l\mathrm{Im}(c_l^m)$ for $m \neq 0$ and $g_l^m = l\mathrm{Re}(c_l^m)$ and $h_l^m = 0$ for $m = 0$.

We insert a spike into $B_r$ at the CMB described by a Fisher–Von Mises probability distribution[51]. This form is chosen because it naturally conforms to the geometry of a spherical surface, and is easily adjusted in angular width and scale. We write

$$B_r^{spike}(c, \theta, \phi) = A\kappa \left(\frac{e^{\kappa \cos \alpha}}{4\pi\sinh(\kappa)}\right) - \frac{A}{4\pi}, \tag{4}$$

where

$$\begin{aligned}\cos\alpha = & \sin\theta\cos\phi\sin\theta^c\cos\phi^c \\ & + \sin\theta\sin\phi\sin\theta^c\sin\phi^c + \cos\theta\cos\theta^c,\end{aligned} \tag{5}$$

is the dot product of the vectors pointing from $r = 0$ to the centre of the spike at $(\theta^c, \phi^c)$ and from $r = 0$ to the point under consideration. The amplitude of the spike is denoted $A$ and the invariance $\kappa$ of the Fisher distribution is related to the angular standard deviation, $\sigma$, by (ref. 51)

$$\kappa = 6,561/\sigma^2. \tag{6}$$

Since we are interested in the limiting case of very thin spikes we have chosen, without loss of generality, a circular spike with equal standard deviations in latitude and longitude, $\sigma = \sigma_\theta = \sigma_\phi$. Multiple spikes can be inserted into $B_r$, each with unique amplitude $A$ and $\sigma$ separated by a distance $\Delta$.

Creating a spike in $B_r$ leaves open the possibility of also creating a monopole field, which violates Maxwell's equations, but this is easily prevented by the introduction of the second term in equation (4). Because the Fisher distribution is a probability density function, the first term in equation (4) integrates to $A$. The second term uniformly distributes field across the core surface to ensure $\int B_r dS = 0$ in accordance with equation (2).

Equation (3) provides a simple means of relating the spectral representations of the field at the surface and CMB. Equation (4) specifies the spike in physical space and so a method is needed to convert between the two. We use the standard spectral transform method[52] with $3/2$Lmax $\theta$ points and $3$Lmax $\phi$ points, which ensures an exact transform between physical and spectral space. The spectral representation also makes it easy to combine two fields together. Since Laplace's

equation is linear, addition of the spike field $B_r^{spike}$ and the observed field $B_r^o$ can be done on each harmonic individually.

We compute the following quantities using the spectral representation: Power spectrum[37]:

$$R_l(r) = (l+1) \sum_{m=0}^{l} \left[ \left( g_l^m \right)^2 + \left( h_l^m \right)^2 \right] \left( \frac{a}{r} \right)^{2l+4}. \quad (7)$$

Minimum Ohmic heating[37]

$$\Psi_m = \left( \frac{4\pi c}{\mu_0^2 \sigma_c} \right) \sum_{l=1}^{Lmax} \sum_{m=0}^{Lmax} \left( \frac{(l+1)(2l+1)(2l+3)}{l} \right) \left[ \left( g_l^m \right)^2 + \left( h_l^m \right)^2 \right]. \quad (8)$$

**Core energy budget.** The energy and entropy equations that encapsulate the gross thermodynamics of core convection and the geodynamo are described in detail elsewhere[38]. The standard model for powering the geodynamo assumes that motion of the core fluid, a mixture of iron plus lighter elements, arises due to cooling and the presence of any radiogenic elements. Cooling leads to freezing of the solid inner core from the centre of the planet because the melting curve is steeper than the ambient temperature profile. As the inner core grows latent heat is released and the light elements partition favourably into the liquid phase, providing a source of gravitational energy. Averaging over a timescale that is long compared to that associated with convection but short compared to the evolutionary timescale of the planet it is assumed that convection mixes the entropy and light elements uniformly throughout the outer core such that the temperature varies adiabatically. The mantle is assumed to be electrically insulating. The global steady state energy equation relates the heat flowing across the CMB, $Q_{cmb}$, to the heat sources inside the core and can be written symbolically as

$$Q_{cmb} = Q_s + Q_L + Q_g + Q_r = A \left( \frac{dT_{cmb}}{dt} \right) + Q_r. \quad (9)$$

Here $Q_s$ is the sensible heat stored in the core, $Q_L$ is the latent heat released as the inner core grows, $Q_g$ is the gravitational energy, $Q_r$ is the heat released by any radiogenic elements and $T_{cmb}$ is the CMB temperature. The quantities $A$ and $B$ represent integrals over core material properties. Equation (9) does not contain magnetic field because magnetic energy is converted locally to heat by Ohmic dissipation. The field does appear in the entropy equation, which can be written symbolically as

$$E_J + E_k = E_s + E_L + E_g + E_r = B \left( \frac{dT_{cmb}}{dt} \right) + E_r \quad (10)$$

where $E_J \geq 0$ is the Ohmic dissipation, $E_k \geq 0$ is the entropy due to thermal conduction, and the terms on the right-hand side are the entropies associated with the heat sources in equation (9). In writing equations (9) and (10) we have neglected small terms representing pressure heating due to core contraction, barodiffusion (pressure-driven diffusion of light elements) and heat of reaction. Viscous dissipation has also been neglected in equation (10); it is expected to be much smaller than the Ohmic dissipation because the kinematic viscosity is about $10^6$ times smaller than the magnetic diffusivity[39] and because magnetic fields tend to suppress small-scale motions[53].

Equations (9) and (10) relate the Ohmic dissipation to the CMB heat flow through the cooling rate at the CMB, $dT_{cmb}/dt$. The quantities $A$ and $B$ are integrals over depth-varying profiles of core material properties[38]. We calculate these quantities using a model[33] that incorporates the Preliminary Reference Earth Model (PREM)[35] density and represents the melting and adiabatic temperature profiles by polynomials. The density jump at the inner core boundary, $\Delta\rho$, determines the amount of light element in the outer core, which in turn affects the melting temperature and the values of all transport properties including the thermal and electrical conductivities, $k$ and $\sigma_c$ respectively. Normal modes give $\Delta\rho = 0.8 \pm 0.2 \, gm \, cc^{-1}$ (ref. 36) and this uncertainty produces a comparable change in $A$ and $B$ to that obtained by combining the uncertainties in all other parameters[34]. We therefore consider changes in $\Delta\rho$ to reflect uncertainty in the calculation and focus on two values, $\Delta\rho = 0.6$ and $0.8 \, gm \, cc^{-1}$, with the corresponding parameters taken from a recent review[34]. Radiogenic heating is not included.

The CMB heat flow is set by mantle convection; it does not necessarily equal the heat conducted out of the core down the adiabatic gradient. Recent reviews[32,38] estimate $Q_{cmb} = 5$–$15$ TW, although $>13$ TW is probably needed at the present day[34,54] in light of the recent upward revision to the core thermal and electrical conductivities[39,55]. The Ohmic dissipation is hard to constrain, partly because the dissipation is thought to occur on small length scales that are not resolved by surface observations[56], and partly because part of the magnetic field is confined within the core. We seek a conservative estimate of the Ohmic dissipation and note that

$$E_J = \int \frac{(\nabla \times B)^2}{\mu_0^2 \sigma T} dV \geq \frac{1}{T_{max}} \int \frac{(\nabla \times B)^2}{\mu_0^2 \sigma} dV = \frac{\Psi}{T_{max}}, \quad (11)$$

where $\mu_0$ is the permeability of free space and $T_{max} \approx 5{,}500$ K is the maximum temperature inside the core[34]. The Ohmic heating $\Psi$ can be calculated directly in numerical geodynamo simulations, although these models are currently restricted

to running in a parameter space far removed from that thought appropriate to Earth's core. Older estimates of this type found $\Psi \approx 0.5$–$1$ TW (refs 28,57), while recent work suggests $\Psi = 3$–$8$ TW (ref. 58). Thermodynamic modelling constrained by output from one particular dynamo simulation[59] yielded the estimate $\Psi \approx 1$–$2$ TW.

The estimates of $\Psi$ above are based on temporal averages of the field as is appropriate for investigating long-term dynamo behaviour. As such they do not contain short-term features like geomagnetic spikes that get averaged out. We therefore assume that these $\Psi$ estimates, denoted $\Psi_D$, represent the Ohmic heating associated with processes acting in the bulk of the core that are responsible for most of the field generation and that we may add to this a contribution $\Psi_m$ due to the geomagnetic spike at the CMB:

$$\Psi = \Psi_D + \Psi_m. \quad (12)$$

As a conservative estimate for $\Psi_m$, we use an analytical solution for the minimum Ohmic heating associated with the observable part of the field given in equation (8) above.

On short timescales the basic assumptions of the model must be re-evaluated. Lateral variations in seismic velocity or density have never been detected[60] and are expected to be small even if the core is not adiabatic throughout[61]. Lateral temperature variations at the top of the core inferred from the observed magnetic field are $10^6$ times smaller than the absolute temperature there[62], backing up the seismic evidence. Seismic models of radial structure agree that the core is very close to adiabatic and well-mixed[35,60], except perhaps in thin layers near the top[63] and bottom[64], but these layers are too thin (O(100) km) to have a significant effect in the calculation. Moreover, numerical models of rotating thermal convection with fixed flux boundaries find that the motion tends to reduce the super-adiabatic temperature difference across the domain as the driving force is increased towards Earth-like values[65]. The assumptions of an adiabatic and well-mixed core therefore seem appropriate when modelling short-term core energetics.

Fluctuations of the fluid velocity, $\mathbf{v}$, that maintains the well-mixed adiabatic state need not average out on short timescales. The CMB is modelled as a simple boundary, that is, a spherical surface with no lateral variations in thermal or electrical properties. These assumptions together with the usual no-slip velocity boundary condition imply that $\mathbf{v}$ enters the governing equations (1) and (2) only through the rate of change of kinetic energy KE, $d(KE)/dt = d\left( 1/2 \int \rho \mathbf{v}^2 dV \right)/dt$, which should be added to the left-hand side of equation (9) and the right-hand side of equation (10). (All $(\mathbf{v} \cdot \nabla)$ terms are transformed to surface integrals that vanish on using the boundary conditions.) The field $\mathbf{B}$ should also be time-dependent and the term $d(ME)/dt = \int \partial \left( \mathbf{B}^2/2\mu_0 \right)/\partial t \, dV$ added to equations (9) and (10), where ME is the magnetic energy. A rough estimate of the magnetic and kinetic energies can be obtained by assuming a constant field $B_0$ and velocity $v_0$. Values of $B_0 = 1$–$10$ mT are chosen; the highest value is 5–10 times present-day estimates for the RMS CMB field strength and four times higher than the core-averaged value inferred from nutations[66]. For $v_0$ the range $10^{-4}$–$10^{-3}$ m s$^{-1}$ is selected, corresponding to roughly 1–10 times the present-day RMS flow speed at the top of the core inferred from geomagnetic secular variation[67]. With these values we obtain $7 \times 10^{19} \leq ME \leq 7 \times 10^{20}$ J and $8 \times 10^{15} \leq KE \leq 8 \times 10^{16}$ J. To make a meaningful contribution to the present-day energy budget of $\approx 13$ TW, $d(ME)/dt$ and $d(KE)/dt$ must contribute at least 1 TW. The timescale $\Delta t$ over which the estimated change in ME and KE would produce 1 TW of power can be estimated from $d(ME)/dt \approx ME/\Delta t_M$ and $d(KE)/dt \approx KE/\Delta t_K$. The required change in kinetic energy would have to occur in $<1$ year, even for flow speeds ten times the present-day CMB value, which seems unlikely. Values of $\Delta t_M$ 20–2,000 years, which also appear unlikely since the largest observed changes in the dipole moment (factor of 3) take 10,000–100,000 years[68].

The spike is not a global feature at the CMB and cannot extend too far into the core without exceeding the available dissipation (Fig. 7). Moreover, RMS field strengths for even the thinnest spikes are comparable to the previous estimates. We therefore conclude that the time-dependent terms make a negligible contribution to the energy budget even in the presence of a spike. Clearly this does not imply that the spike itself is a static feature of the field.

**Data availability.** The data that support the findings of this study are available from the corresponding author upon reasonable request.

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

## Acknowledgements

C.D. is supported by Natural Environment Research Council independent research fellowship NE/L011328/1 and a Green scholarship at IGPP. C.C. acknowledges support from NSF grants EAR-1246826 and EAR-1623786.

## Author contributions

C.D. and C.C. designed the project and wrote the paper. C.D. developed the numerical model of the spike and the core energy budget. C.C. developed the error analysis applied to the paleointensity data.

## Additional information

**Competing interests:** The authors declare no competing financial interests.

**Publisher's note**: 

