## [Peer Review File · Nature Communications]

Reviewers' Comments:

Reviewer #1 (Remarks to the Author)

The paper models the very high values of paleomagnetic intensity inferred from two samples from Israel and Jordan for the time around 1000 B.C. in terms of a concentrated very high-intensity magnetic flux spot at the core-mantle boundary below that region. Similar magnetic features at the top of the core have not been observed in the historical (past 400 yr) magnetic field and it would be very interesting if their existence could be substantiated. However, the evidence presented in the paper is insufficient to support the claim.

As is demonstrated in the paper and is obvious in a qualitative sense, even a very concentrated magnetic spike at the core-mantle boundary must have an expression at the Earth's surface that extends over a wide region (Fig. 1). While the two samples from Israel and Jordan show very high-intensity values, other samples from nearby locations indicate normal intensities. There is no evidence for a smooth fall-off of intensity with distance. The model significantly overpredicts intensities at locations roughly 40 degrees away, in India and in Mali (Fig. 1 bottom).

The data quality is not discussed in detail (and no error bars are given), but the set is called "heterogeneous" and "with minimal quality controls" (page 4). Unless it can be shown that the two high-intensity samples from Israel and Jordan are of high quality, the lack of evidence for a broad high-intensity region on the Earth's surface may simply indicate a problem with these two samples.

No directional data are discussed. The hypothetical extreme field spike at the core mantle boundary would not only raise the intensity at the surface, but must also lead to rather strong declination and inclination anomalies in a wide area. Assuming that directional data for this epoch and wider region exist, the case for the spike would be strengthened if it could be shown that inclination and declination are consistent with those predicted by the spike hypothesis. Without such additional support, the interpretation is on too weak a footing.

Reviewer #2 (Remarks to the Author)

Review to "Geomagnetic spikes on the core-mantle boundary":

This paper is well motivated by the enigmatic origin of the strongly localized (in space and time) spike of geomagnetic field intensity in Israel/Jordan around 1000BC. The authors superpose a localized structure on an archeomagnetic field model and fit its center, amplitude and width. Then they use core energy budget arguments to further constrain the CMB spike. Because during the spike's period the dipole was increasing, and because equator crossing is unlikely for rapidly rotating flows, they speculate that the spike drifted northward. In addition, the authors argue that strong diffusion effects, which are likely for such a sharp structure, must have diminished the spike.

The idea of combining geometric and energetic constraints to fit the spike is interesting. The main achievement of the combined geometric and energetic constraints is the assessment of the size and amplitude of the CMB patch that was responsible for the spike. No solution is proposed for the temporal localization. Some kinematic scenarios are proposed. The paper is original, interesting and worth publication. However, whether it is worth publication in Nature Communications - I am not sure.

My main comments are:

A. Spike best model: I fully understand that it is impossible to simultaneously fit the high

intensities from Israel/Jordan with the low intensities from Syria/Cyprus/Egypt (Fig. 1b). However, the poor fit seems to go beyond the Levant. In Fig. 1a the colors of symbols seem rather different from the global colors, e.g. red point in the blue Pacific, green and blue points in orange Europe, red and blue points on the same orange contour in China and India - is this a reasonable fit? Also it would be nice to add an intensity map from a present day field model to highlight the spike-like nature in the 1000BC map.

B. Geometric constraint: Why did you center Fig. 2 away from Jordan/Israel? It is expected that the spike will be characterized by a high degree on the CMB, but does this high degree power exceed the dipole power on the CMB? Would be interesting to plot the power of $Br_0 + Br_{\text{spike}}$ on the CMB for the best model.

C. Temporal localization: You don't really have an answer for that, you just say that either there are severe dating errors with the Syria/Cyprus/Egypt data, or the spike moved super fast but the latter is very unlikely. So the good question remains unanswered...

D. Consistency with present-day gyre: The maps of present-day core flow models you refer to (Finlay et al., 2015) indeed show a large-scale gyre with meridional flow that may carry flux to different latitudes and by that change the axial dipole. However, in the Levant these present-day core flow models predict equatorward motion, not poleward as you argue. This has to be corrected. In fact these are good news for you. Today the gyre is anti-cyclonic, moving normal flux equatorward in the Levant, by that reducing the dipole. In 1000BC according to your scenario normal flux drifted poleward in the Levant, by that increasing the dipole.

E. Diffusion: Good point, diffusion is likely strong for such a localized structure, but how would you incorporate it in a dynamical model? Diffusion is already incorporated in numerical dynamos, and Earth-like magnetic Reynolds numbers are accessible. Worth citing Amit and Christensen (2008) that inferred diffusion SV from numerical dynamos to improve core flow inversions. They also noted that effectively diffusion plays a greater role at the CMB than what may be expected based on conventional estimates of the magnetic Reynolds number.

Detailed comments:

1. Abstract line 7: Equatorial dipole always spans 180 degrees, surely you don't mean that. Rephrase "characterized by a low-latitude intensity peak that spans ...".

2. Page 1 first paragraph: Christensen et al. (2010) reproduced the geomagnetic field morphology without heterogeneous boundary conditions and without waves (as you well know, recall Davies and Constable, 2014). High-latitude intense flux patches can easily be reproduced in numerical dynamos without heterogeneous CMB due to the tangent cylinder effect (e.g. Christensen et al., 1998). In homogeneous CMB conditions these patches may reside in any longitude, whereas with heterogeneous CMB preferred locations of these flux patches arise (e.g. Olson and Christensen, 2002). Core flow motions may constrain the SV more than the field morphology. In the last sentence the word NOW suggests that there is some new data base in 2016 that allows to better address the question of spikes, while Holocene field models were around for a while. Please rewrite this paragraph more accurately.

3. Caption Fig. 1: Add "(a)" and "(b)" at top and bottom (you refer to Fig. 1b in the text). Add that (a) is at Earth's surface. Add that σ is standard deviation of the spike (i.e. width). You describe dashed horizontal lines, I see only one dashed horizontal line...

4. Fig. 2: Change blue titles to "Harmonic degree of max energy at surface" (not to be confused with truncation degree).

5. Page 3 last sentence: Counter-intuitive? Please elaborate.

6. Page 4 line 4 from bottom: Obviously Br on the CMB can best be calculated from geomagnetic field models, why do we need these dynamical models to obtain RMS Br on the CMB? Do you mean internal field RMS? $|B|$ on the CMB (including toroidal field just below)? Small-scale Br beyond $\ell > 14$? Gillet et al. (2010) for example infer the toroidal field inside the core. Please clarify.

7. Page 6 last paragraph: Please cite Olson and Amit (2006) in the context of dipole decrease/increase due to normal flux migration to lower/higher latitudes.

8. Methods until equation 4: Textbook, can be deleted.

References:

- * Amit, H., Christensen, U., 2008. Accounting for magnetic diffusion in core flow inversions from geomagnetic secular variation. *Geophys. J. Int.*, 175, 913-924.
- * Christensen, U., Aubert, J., Hulot, G., 2010. Conditions for Earth-like geodynamo models. *Earth planet. Sci. Lett.*, 296, 487-496.
- * Christensen, U., Olson, P., Glatzmaier, G., 1998. A dynamo model interpretation of geomagnetic field structures. *Geophys. Res. Lett.*, 25, 1565-1568.
- * Davies, C. J., Constable, C. G., 2014. Insights from geodynamo simulations into long-term geomagnetic field behaviour. *Earth planet. Sci. Lett.*, 404, 238-249.
- * Finlay, C. C., Aubert, J., Gillet, N., 2016. Gyre-driven decay of the Earth's magnetic dipole. *Nature Commun.* 7, 1-8.
- * Gillet, N., Jault, D., Canet, E., Fournier, A., 2010. Fast torsional waves and strong magnetic field within the Earth's core. *Nature* 465, 74-77.
- * Olson, P., Amit, H., 2006. Changes in Earth's dipole. *Naturwissenschaften*, 93, 11, 519-542.
- * Olson, P., Christensen, U., 2002. The time averaged magnetic field in numerical dynamos with nonuniform boundary heat flow. *Geophys. J. Int.*, 151, 809-823.

I recommend possible publication after major revisions.

Best regards,

Hagay Amit

Reviewer #3 (Remarks to the Author)

Geomagnetic spikes are intriguing features of the geomagnetic secular variation, that have recently come to the fore after their discovery by Ben Yosef et al. (EPSL, 2009) and Shaar et al. (EPSL, 2011). Analysis of the magnetic properties of Near East copper slag residues by these authors points to a very high geomagnetic intensity around 1000 BC in this region. Even more striking is the fact that this intensity is maintained during a very brief time span, suggesting rates of change of intensity several tens of times larger than what is observed at present-day. Such spikes challenge our current understanding of the geodynamo, the process that generates and sustains the geomagnetic field against Ohmic decay in the Earth's core. They deserve the observers' and theoreticians' attention. They already received some: on the theoretical side, Livermore et al. (EPSL, 2014) sought core flows optimized in order to account for the rapid changes suggested by spikes. Their conclusion was that the energy required for such changes to be possible was beyond commonly accepted values, by a factor of 5 or so. Building on this optimized core flow approach, Fournier et al. (GRL, 2015) constructed two timedependent global spikes models (one reasonable, that does not account for the observed variations, and one optimistic, that does so, allowing for the 5 fold increase in energy budget already discussed). They used these models to see whether such spikes could have a measurable imprint on the production of cosmogenic ^{14}C and ^{10}Be in the Earth's atmosphere. The imprint computed in the optimistic setup did not agree with the cosmogenic data, unless the ages of the spikes were allowed to be shifted by a few decades. That study showed again the difficulty to reconcile spikes (if one admits their existence) with our current understanding of core dynamics. It is more the overall timing of these events that is difficult to comprehend, rather than the amplitude of the intensity itself. Note that both Livermore et al. and Fournier et al. ignored the effect of radial diffusion in their analysis, which may be a bit reckless if spikes originate from upwellings underneath the core-mantle boundary (the effect of lateral diffusion was shown to be negligible). Even more intriguing is the recent paper by Bourne et al. (EPSL, 2016) whose study of sediments in Hall's Cave, Texas, points to an intensity spike that could possibly be coeval with one of the Levantine spikes,

while reflecting a longer-lasting intensity high, with time variations more in line with what we think is commonly acceptable. This study (which has appeared after this manuscript was submitted) is particularly relevant to the work by C. Davies and C. Constable, since their main focus is not on the time variations associated with the Levantine events, but on their spatial extent. The 'spike' terminology initially referred to both the spatial and temporal variations associated with these events (the latter being the most intriguing), but the spatial extent is the only one of interest for the authors, who do not consider any time-dependency in their analysis. Davies and Constable consider a background magnetic field at the core-mantle boundary (CMB) provided by a time-dependent, global magnetic field model covering the Holocene. Given this background field (which fails to account for the Levantine spikes), they place an extra amount of magnetic flux at the core surface, of limited spatial extent, and consider how such a parameterized and adjustable patch affects geomagnetic intensity at the Earth's surface, in particular in the Near East. An optimization based on the available data ensues, and an ensemble of models of CMB Br-spike is proposed, whose typical amplitude is not in contradiction with independent estimates on the maximum amplitude of Br at the core surface. By virtue of geometric attenuation within the mantle, the surface signature of the CMB Br-spikes has always a rather large extent and is in fact dominated by the dipole (SH degree $\ell = 1$). It is also found that the energetics associated with such events are not incompatible with global core energetics. All in all that is an interesting study, but further work is needed for publication to be within sight. I list several points below. Some of these reflect the fact that the recent literature should be discussed in a little bit more detail.

1. The Bourne et al. paper is clearly of importance for the study conducted here, for it would require the existence of two intensity highs separated by 11 000 km at the Earth's surface. This point should be discussed. As a matter of fact, other data (one seems to be in Hawaii, Fig. 1, top) reflect high values not accounted for by the model. So it seems that the solution to this would be to place a series of spikes at the surface of the core that could produce highs in intensity compatible with the various observations. But since these spike models lead predominantly to $\ell = 1$ changes at the surface, would not they give rise to unrealistically large values of the VADM? Or would their effect cancel each other? In their analysis, Livermore et al. studied the dual spike problem and concluded that, for a fixed energy budget, the occurrence of multiple spikes was not a favorable factor.
2. The current figure 2 is not easy to understand. I'd rather see Lowes-Mauersberger spectra with and without the spike, at the surface of the core (eg Jackson 2003, his figure 2), and at the surface of the Earth, for a few representative spikes models. This would illustrate nicely the effect of geometric attenuation within the mantle.
3. For a suitable spike model, what is the value of the cylindrical radius characterizing the eccentric dipole? How does it compare with the value of several hundred km computed by Fournier et al. (their Figure 2a)? The eccentricity is a concept useful to visualize the effect of the spike at the Earth's surface. On longer time scales, it is thought to reflect changes in core flow connected with inner core growth (Olson & Deguen, Nature Geoscience, 2012).
4. A scenario is put forward regarding how the CMB feature associated with the spike may have contributed to the dipole high around that epoch, after its initial birth in the equator and its subsequent northward advection by core flow (in particular a large scale gyre). A recent paper by Finlay et al. (Nature Comm, 2016), which must also have appeared after the submission of this manuscript provides a gyre-based scenario for dipole decay which states the opposite, namely that dipole decay occurs via poleward transport of reverse polarity flux and equatorward transport of normal polarity flux. I would like the authors to discuss this apparent contradiction in detail.
5. On the data side: the misfit of $\sim 20 \mu\text{T}$ obtained by the authors is said to be reasonable given the heterogeneous intensity data with "minimal quality controls": what is the

meaning of this last statement? Should not the authors only retain those data that they estimate are appropriate for their analysis? 6. On the same note: the authors should work under uncertainty and compute L1 and L2 based on the uncertainty estimate for each datum. Minor points 1. 2800 km below Earth's surface → 2900 km 2. Methods, Eqs. (4) and (5): The $\cos \theta$ should be a $\cos \alpha$ or some other Greek letter (the angular distance between the spike center and the observation point)

Response to Reviewers

We thank the reviewers for their constructive comments which are addressed in our responses below (responses in black).

Reviewer #1 (Remarks to the Author):

The paper models the very high values of paleomagnetic intensity inferred from two samples from Israel and Jordan for the time around 1000 B.C. in terms of a concentrated very high-intensity magnetic flux spot at the core-mantle boundary below that region. Similar magnetic features at the top of the core have not been observed in the historical (past 400 yr) magnetic field and it would be very interesting if their existence could be substantiated. However, the evidence presented in the paper is insufficient to support the claim.

Response: Our intention is not to necessarily support the claim of a spike, but rather to explore the implications of a high intensity magnetic flux patch residing on the core-mantle boundary. We realize that the original manuscript did not make this clear. We have tried to make this point clearer in the abstract (lines 12-13) and presented an expanded discussion of the data, which is detailed below.

It is true that magnetic features as strong as the Levantine spike have not been seen for the past 400 years, but the field is currently weakening. Associated with this we do see a train of mobile high intensity flux patches south of the equator (e.g., Jackson, 2003). In this context high intensity means $B_r \sim 1$ mT, corresponding to a broad spike width at the CMB of $35 - 40^\circ$ (see Figure 4). The regularized core field inversions used for modern field modeling cannot support the resolution of anything more localized, because of masking by the crustal field. It is possible that the Levantine data represent a similar (but much stronger) phenomenon in the northern hemisphere at a time when the field is growing in strength.

As is demonstrated in the paper and is obvious in a qualitative sense, even a very concentrated magnetic spike at the core-mantle boundary must have an expression at the Earth's surface that extends over a wide region (Fig. 1). While the two samples from Israel and Jordan show very high-intensity values, other samples from nearby locations indicate normal intensities. There is no evidence for a smooth fall-off of intensity with distance. The model significantly overpredicts intensities at locations roughly 40 degrees away, in India and in Mali (Fig. 1 bottom).

The data quality is not discussed in detail (and no error bars are given), but the set is called "heterogeneous" and "with minimal quality controls" (page 4). Unless it can be shown that the two high-intensity samples from Israel and Jordan are of high quality, the lack of evidence for a broad high-intensity region on the Earth's surface may simply indicate a problem with these two samples.

Response: The available data used to produce a spatial map of the Levantine spike are heterogeneous in the sense that they come from a variety of archeological and

geological materials, via differing laboratory protocols, and with a wide range of age controls and documentation. In view of these issues, and the lengthy analyses that are available in the literature (Tauxe & Yamazaki 2015), we wished to avoid a discussion of the data quality. We have revised section 1 of the Supplementary Information and moved this into the main text to facilitate an expanded discussion of available data, and allow readers to appreciate the number of data available and their error bars (Lines 33-99 and Figures 1-2 of the revised text).

The high quality and reproducibility of the Levantine data has been demonstrated in a recent publication (Shaar et al. 2016). Evidence for high intensities in the region around 1000BC is also supported by data from Georgia (Shaar et al. 2013) and Turkey (Ertepinar et al. 2012). These references have been added to the discussion on lines 33-99 of the revised manuscript.

No directional data are discussed. The hypothetical extreme field spike at the core mantle boundary would not only raise the intensity at the surface, but must also lead to rather strong declination and inclination anomalies in a wide area. Assuming that directional data for this epoch and wider region exist, the case for the spike would be strengthened if it could be shown that inclination and declination are consistent with those predicted by the spike hypothesis. Without such additional support, the interpretation is on too weak a footing.

Response: The reviewer raises an important issue about the kind of data available for testing consistency of any spike signal. Unfortunately there are very few archeomagnetic direction data in the relevant time interval – only 2 results were available in the Geomag database between 1150 and 850 BC in the geographic bin represented in the new Figure 1(e), and this is clearly a significant limitation. More directional data are needed, although since our original submission several have been published for both the Levantine and Texas locations. These are now discussed on lines 235-244 of the revised text.

In the revised manuscript we now consider the case of two spikes, one under the Levant and another under Texas (see Comment 1 by Reviewer 3). Global maps of F , I and D for this case are presented in Supplementary Figure 5 and are discussed on lines 68-81 of the Supplementary Information. We find that the model predictions are compatible with the limited available Turkish, Levantine and North American observations.

Reviewer #2 (Remarks to the Author):

Review to "Geomagnetic spikes on the core-mantle boundary":

This paper is well motivated by the enigmatic origin of the strongly localized (in space and time) spike of geomagnetic field intensity in Israel/Jordan around 1000BC. The authors superpose a localized structure on an archeomagnetic field model and fit its center, amplitude and width. Then they use core energy budget arguments to further constrain the CMB spike. Because during the spike's period the dipole was increasing, and because equator crossing is unlikely for rapidly rotating

flows, they speculate that the spike drifted northward. In addition, the authors argue that strong diffusion effects, which are likely for such a sharp structure, must have diminished the spike.

The idea of combining geometric and energetic constraints to fit the spike is interesting. The main achievement of the combined geometric and energetic constraints is the assessment of the size and amplitude of the CMB patch that was responsible for the spike. No solution is proposed for the temporal localization. Some kinematic scenarios are proposed. The paper is original, interesting and worth publication. However, whether it is worth publication in Nature Communications - I am not sure.

My main comments are:

- A. Spike best model: I fully understand that it is impossible to simultaneously fit the high intensities from Israel/Jordan with the low intensities from Syria/Cyprus/Egypt (Fig. 1b). However, the poor fit seems to go beyond the Levant. In Fig. 1a the colors of symbols seem rather different from the global colors, e.g. red point in the blue Pacific, green and blue points in orange Europe, red and blue points on the same orange contour in China and India - is this a reasonable fit? Also it would be nice to add an intensity map from a present day field model to highlight the spike-like nature in the 1000BC map.

Response: We thank the reviewer for making this important observation. Indeed, the original manuscript did not make clear what could be expected as a reasonable fit between data and model. We have added to the revised manuscript an intensity map from CALS10k.2 at 1000 BC overlaid with the available data (new Figure 3b). This shows that many of the discrepancies pointed out by the reviewer (in particular Hawaii, Switzerland, China and India) arise due to a misfit between the data and the global field model and are not caused by the addition of the spike field. It is also worth noting that some of the data were not recorded exactly at 1000 BC and are colour-coded in $10\mu T$ bins. We have expanded the discussion of misfit starting on line 198. We have also added the intensity map for the present day (new Figure 3a) and discussed it on line 99.

- B. Geometric constraint: Why did you center Fig. 2 away from Jordan/Israel? It is expected that the spike will be characterized by a high degree on the CMB, but does this high degree power exceed the dipole power on the CMB? Would be interesting to plot the power of $Br_0 + Br_{\text{spike}}$ on the CMB for the best model.

Response: There is no significance to using the equator for the calculations in Figure 2; this location simply makes it easy to calculate the longitudinal width of the spike and to visualize the results. Since there is no background field this suite of calculations could have been performed with spikes centered in any location. Depending on the values of A and σ it is indeed possible for the power at high degree to exceed the dipole power. Power spectra of the spike field, background

field and total field at both the CMB and Earth's surface are now shown in the new Figure 5 and discussed on lines 158 and 328.

C. Temporal localization: You don't really have an answer for that, you just say that either there are severe dating errors with the Syria/Cyprus/Egypt data, or the spike moved super fast but the latter is very unlikely. So the good question remains unanswered...

Response: The temporal behavior of the Levantine spike is indeed a complex issue, in large part because it has not been clear how to characterize the spike. By focusing on spatial structure we have obtained new constraints on the CMB spike location, geometry, and amplitude, all of which feed into models of the temporal behavior (Livermore et al. 2014). In particular, we have shown that the spike is not well-resolved in present archeomagnetic field models and that the power spectrum and dissipation in plausible model spikes is very different to those used when the spike geometry was not explicitly considered (Livermore et al. 2014). Moreover, our study strongly suggests that radial diffusion, which has previously been ignored, must be accounted for in models of the temporal evolution. These results will enable future studies to focus on the complex issues that are specific to temporal evolution such as the lack of adequate age constraints in the present database and the difficulty in temporal matching at globally distributed locations. We have discussed these issues starting on line 325 of the revised manuscript.

D. Consistency with present-day gyre: The maps of present-day core flow models you refer to (Finlay et al. 2016) indeed show a large-scale gyre with meridional flow that may carry flux to different latitudes and by that change the axial dipole. However, in the Levant these present-day core flow models predict equatorward motion, not poleward as you argue. This has to be corrected. In fact these are good news for you. Today the gyre is anti-cyclonic, moving normal flux equatorward in the Levant, by that reducing the dipole. In 1000BC according to your scenario normal flux drifted poleward in the Levant, by that increasing the dipole.

Response: We are very grateful to the reviewer for this comment. The scenario we envisage (which was not clearly stated in the original manuscript) is that the Levantine flux patch followed the pattern of the current anticyclonic gyre, perhaps even with a train of high flux patches moving westward somewhat north of the equator, ultimately moving north to join flux lobes near the tangent cylinder. We note that the intensity spike identified at Hall's Cave (Bourne et al. 2016), interpreted as an intense flux spot on the CMB, would also enhance the dipole, since flux under North America is located near the northward leg of the anticyclonic gyre and so would also be swept towards the pole. These issues are discussed in the paragraph beginning on line 312 of the revised text.

E. Diffusion: Good point, diffusion is likely strong for such a localized structure, but how would you incorporate it in a dynamical model? Diffusion is already incorporated in numerical dynamos, and Earth-like magnetic Reynolds numbers are

accessible. Worth citing (Amit & Christensen 2008) that inferred diffusion SV from numerical dynamos to improve core flow inversions. They also noted that effectively diffusion plays a greater role at the CMB than what may be expected based on conventional estimates of the magnetic Reynolds number.

Response: We thank the reviewer for pointing out the work of Amit & Christensen. Since we have not constructed a dynamical model we do not wish to speculate on the precise role of radial diffusion. In the revised manuscript (beginning on line 337) we have noted that the effects of radial diffusion could be incorporated into a core flow model of the spike via a parameterization as suggested by Amit & Christensen (2008).

Detailed comments:

1. Abstract line 7: Equatorial dipole always spans 180 degrees, surely you don't mean that. Rephrase "characterized by a low-latitude intensity peak that spans ...".

Response: We thank the reviewer for this comment and have made altered the text accordingly.

2. Page 1 first paragraph: (Christensen et al. 2010) reproduced the geomagnetic field morphology without heterogeneous boundary conditions and without waves (as you well know, recall (Davies & Constable 2014)). High-latitude intense flux patches can easily be reproduced in numerical dynamos without heterogeneous CMB due to the tangent cylinder effect (Christensen et al. 1998). In homogeneous CMB conditions these patches may reside in any longitude, whereas with heterogeneous CMB preferred locations of these flux patches arise (Olson & Christensen 2002). Core flow motions may constrain the SV more than the field morphology. In the last sentence the word NOW suggests that there is some new data base in 2016 that allows to better address the question of spikes, while Holocene field models were around for a while. Please rewrite this paragraph more accurately.

Response: We thank the reviewer for this comment and have rewritten the paragraph (lines 24-31).

3. Caption Fig. 1: Add "(a)" and "(b)" at top and bottom (you refer to Fig. 1b in the text). Add that (a) is at Earth's surface. Add that σ is standard deviation of the spike (i.e. width). You describe dashed horizontal lines, I see only one dashed horizontal line...

Response: Figure 1 is now Figure 3. It has been modified and includes the requested changes.

4. Fig. 2: Change blue titles to "Harmonic degree of max energy at surface" (not to be confused with truncation degree).

Response: We have made the required changes.

5. Page 3 last sentence: Counter-intuitive? Please elaborate.

Response: Power in the smaller scales of the spike field is significantly attenuated between the core-mantle boundary and the surface. Indeed, all of our spike models (with no background field) show that the harmonic degree with most power at Earth's surface is $l = 1$. This point is hopefully clearer now that plots of the power spectra and the CMB and Earth's surface have been added to the revised manuscript (new Figure 5).

6. Page 4 line 4 from bottom: Obviously B_r on the CMB can best be calculated from geomagnetic field models, why do we need these dynamical models to obtain RMS B_r on the CMB? Do you mean internal field RMS? $|B|$ on the CMB (including toroidal field just below)? Small-scale B_r beyond $l > 14$? (Gillet et al. 2010) for example infer the toroidal field inside the core. Please clarify.

Response: We thank the reviewer for raising this point. To clarify, the point is that the RMS CMB field strength could be greater than that inferred from field models if there is substantial power in scales that are too small to resolve, like those characterizing our model spikes. However, when quoting values of B_{RMS} from the literature we have conflated estimates of the internal field strength with the RMS CMB field strength. This has now been corrected on lines 225-233 and makes a minor difference ($\sim 1^\circ$) to the allowable values of σ ; it does not change the conclusions.

7. Page 6 last paragraph: Please cite (Olson & Amit 2006) in the context of dipole decrease/increase due to normal flux migration to lower/higher latitudes.

Response: We have made the required change.

8. Methods until equation 4: Textbook, can be deleted.

Response: We prefer to leave this intact for the broader paleomagnetic audience.

Reviewer #3 (Remarks to the Author):

Geomagnetic spikes on the core-mantle boundary by Christopher Davies and Catherine Constable

Geomagnetic spikes are intriguing features of the geomagnetic secular variation, that have recently come to the fore after their discovery by Ben Yosef et al. (EPSL, 2009) and (Shaar et al. 2011). Analysis of the magnetic properties of Near East copper slag residues by these authors points to a very high geomagnetic intensity around 1000 BC in this region. Even more striking is the fact that this intensity is

maintained during a very brief time span, suggesting rates of change of intensity several tens of times larger than what is observed at present-day.

Such spikes challenge our current understanding of the geodynamo, the process that generates and sustains the geomagnetic field against Ohmic decay in the Earth's core. They deserve the observers' and theoreticians' attention.

They already received some: on the theoretical side, (Livermore et al. 2014) sought core flows optimized in order to account for the rapid changes suggested by spikes. Their conclusion was that the energy required for such changes to be possible was beyond commonly accepted values, by a factor of 5 or so. Building on this optimized core flow approach, (Fournier et al. 2015) constructed two time dependent global spikes models (one reasonable, that does not account for the observed variations, and one optimistic, that does so, allowing for the 5 fold increase in energy budget already discussed). They used these models to see whether such spikes could have a measurable imprint on the production of cosmogenic ^{14}C and ^{10}Be in the Earth's atmosphere. The imprint computed in the optimistic setup did not agree with the cosmogenic data, unless the ages of the spikes were allowed to be shifted by a few decades. That study showed again the difficulty to reconcile spikes (if one admits their existence) with our current understanding of core dynamics. It is more the overall timing of these events that is difficult to comprehend, rather than the amplitude of the intensity itself. Note that both (Livermore et al. 2014) and (Fournier et al. 2015) ignored the effect of radial diffusion in their analysis, which may be a bit reckless if spikes originate from upwellings underneath the core-mantle boundary (the effect of lateral diffusion was shown to be negligible).

Even more intriguing is the recent paper by (Bourne et al. 2016) whose study of sediments in Hall's Cave, Texas, points to an intensity spike that could possibly be coeval with one of the Levantine spikes, while reflecting a longer-lasting intensity high, with time variations more in line with what we think is commonly acceptable.

This study (which has appeared after this manuscript was submitted) is particularly relevant to the work by C. Davies and C. Constable, since their main focus is not on the time variations associated with the Levantine events, but on their spatial extent. The 'spike' terminology initially referred to both the spatial and temporal variations associated with these events (the latter being the most intriguing), but the spatial extent is the only one of interest for the authors, who do not consider any time-dependency in their analysis.

Davies and Constable consider a background magnetic field at the core-mantle boundary (CMB) provided by a time-dependent, global magnetic field model covering the Holocene. Given this background field (which fails to account for the Levantine spikes), they place an extra amount of magnetic flux at the core surface, of limited spatial extent, and consider how such a parameterized and adjustable patch affects geomagnetic intensity at the Earth's surface, in particular in the Near East. An optimization based on the available data ensues, and an ensemble of models of CMB

B_r -spike is proposed, whose typical amplitude is not in contradiction with independent estimates on the maximum amplitude of B_r at the core surface.

By virtue of geometric attenuation within the mantle, the surface signature of the CMB B_r -spikes has always a rather large extent and is in fact dominated by the dipole (SH degree $l = 1$). It is also found that the energetics associated with such events are not incompatible with global core energetics. All in all that is an interesting study, but further work is needed for publication to be within sight. I list several points below. Some of these reflect the fact that the recent literature should be discussed in a little bit more detail.

1. The (Bourne et al. 2016) paper is clearly of importance for the study conducted here, for it would require the existence of two intensity highs separated by 11 000 km at the Earth's surface. This point should be discussed. As a matter of fact, other data (one seems to be in Hawaii, Fig. 1, top) reflect high values not accounted for by the model. So it seems that the solution to this would be to place a series of spikes at the surface of the core that could produce highs in intensity compatible with the various observations. But since these spike models lead predominantly to $l = 1$ changes at the surface, would not they give rise to unrealistically large values of the VADM? Or would their effect cancel each other? In their analysis, (Livermore et al. 2014) studied the dual spike problem and concluded that, for a fixed energy budget, the occurrence of multiple spikes was not a favorable factor.

Response: We thank the reviewer for bringing the Bourne paper to our attention. The issue of multiple spikes was considered in the Supplementary Information accompanying the original manuscript and we have now extended this discussion to include the specific case of two spikes, one under Texas and one under the Levant. We show that it is possible to find two-spike models with lower L^1 than one-spike models, which is not surprising because of the additional degrees of freedom (amplitude and width of the second spike), and acceptable fits to recently obtained directional data. Adding a spike makes a small change to the Ohmic dissipation as long as the spike is not too thin and increases the VADM by $\sim 10\%$, which is compatible with the available data around 1000 BC. Clearly it is possible to produce an unacceptably high VADM and Ohmic dissipation by adding more strong spikes, but there is little paleomagnetic evidence for this scenario and we do not investigate it further. In the revised manuscript we have added a new figure and paragraph to the Supplementary Information starting on line 68, and a new paragraph to the main text starting on line 279.

2. The current figure 2 is not easy to understand. I'd rather see Lowes-Mauersberger spectra with and without the spike, at the surface of the core (eg (Jackson 2003), his figure 2), and at the surface of the Earth, for a few representative spikes models. This would illustrate nicely the effect of geometric attenuation within the mantle.

Response: We think that Figure 2 (now Figure 4) provides a succinct visual representation of the spike width results, but agree that it is certainly easier to

visualize the spike using the power spectrum R_l . However, we have found that plots of R_l become messy when more than a couple of spectra are plotted on top of each other. As a compromise we have retained the old Figure 2 (now Figure 4) and added the new Figure 5, which shows R_l for the spike and background field at the CMB and Earth's surface for one of the low-misfit models.

3. For a suitable spike model, what is the value of the cylindrical radius characterizing the eccentric dipole? How does it compare with the value of several hundred km computed by Fournier et al. (2015) (their Figure 2a)? The eccentricity is a concept useful to visualize the effect of the spike at the Earth's surface. On longer time scales, it is thought to reflect changes in core flow connected with inner core growth (Olson & Deguen 2012).

Response: In order to minimize the number of figures we have instead calculated the dipole tilt and used this to annotate Figure 3 with the best-fitting dipole axis. Fournier et al. (2015) also calculated dipole tilt and we obtain slightly lower values than their extreme model.

4. A scenario is put forward regarding how the CMB feature associated with the spike may have contributed to the dipole high around that epoch, after its initial birth in the equator and its subsequent northward advection by core flow (in particular a large scale gyre). A recent paper by (Finlay et al. 2016), which must also have appeared after the submission of this manuscript provides a gyre-based scenario for dipole decay which states the opposite, namely that dipole decay occurs via poleward transport of reverse polarity flux and equatorward transport of normal polarity flux. I would like the authors to discuss this apparent contradiction in detail.

Response: We thank the reviewer for raising this point, which was also raised by Reviewer 2. Please see the discussion in response to Reviewer 2 point D above.

5. On the data side: the misfit of $\sim 20 \mu\text{T}$ obtained by the authors is said to be reasonable given the heterogeneous intensity data with "minimal quality controls": what is the meaning of this last statement? Should not the authors only retain those data that they estimate are appropriate for their analysis?

Response: We thank the reviewer for this comment, which is similar to comment 2 of Reviewer 1 and comment A of Reviewer 2. Please see the responses to these comments above.

6. On the same note: the authors should work under uncertainty and compute L1 and L2 based on the uncertainty estimate for each datum.

Response: See response to point 5 – We can cite L1 and L2 misfits, but it is difficult to put the values in a rational context because in general paleointensity estimates do not have good controls on uncertainty.

Minor points

1. 2800 km below Earth's surface \rightarrow 2900 km

Response: We have made the required change.

2. Methods, Eqs. (4) and (5): The $\cos \theta$ should be a $\cos \alpha$ or some other Greek letter (the angular distance between the spike center and the observation point)

Response: We thank the reviewer for this comment and have made the suggested change.

References

- Amit, H. & Christensen, U.R., 2008. Accounting for magnetic diffusion in core flow inversions from geomagnetic secular variation. *Geophysical Journal International*, 175(3), pp.913–924.
- Bourne, M.D. et al., 2016. High-intensity geomagnetic field “ spike ” observed at ca . 3000 cal BP in Texas , USA. *Earth and Planetary Science Letters*, 442, pp.80–92. Available at: <http://dx.doi.org/10.1016/j.epsl.2016.02.051>.
- Christensen, U., Olson, P. & Glatzmaier, G.A., 1998. A dynamo model interpretation of geomagnetic field structures. *Geophysical Research Letters*, 25(10), pp.1565–1568. Available at: [papers3://publication/uuid/658C68E9-98E2-4F03-A9CC-2EB7922E4E25](https://pubs.gsa.org/publication/uuid/658C68E9-98E2-4F03-A9CC-2EB7922E4E25).
- Christensen, U.R., Aubert, J. & Hulot, G., 2010. Conditions for Earth-like geodynamo models. *Earth Planet. Sci. Lett.*, 296, pp.487–496.
- Davies, C.J. & Constable, C.G., 2014. Insights from geodynamo simulations into long-term geomagnetic field behaviour. *Earth Planet. Sci. Lett.*, 404, pp.238–249.
- Ertepinar, P. et al., 2012. Archaeomagnetic study of five mounds from Upper Mesopotamia between 2500 and 700 BCE: Further evidence for an extremely strong geomagnetic field ca. 3000 years ago. *Earth and Planetary Science Letters*, 357-358, pp.84–98.
- Finlay, C.C., Aubert, J. & Gillet, N., 2016. Gyre-driven decay of the Earth’s magnetic dipole. *Nat. Commun*, 7, pp.1–8. Available at: <http://dx.doi.org/10.1038/ncomms10422>.
- Fournier, A. et al., 2015. The impact of geomagnetic spikes on the production rates of cosmogenic ¹⁴C and ¹⁰Be in the Earth’s atmosphere. *Geophys. Res. Lett.*, 42, pp.2759–2766.
- Gillet, N. et al., 2010. Fast torsional waves and strong magnetic field within the Earth’s core. *Nature*, 465, pp.74–77.
- Jackson, A., 2003. Intense equatorial flux spots on the surface of the Earth’s core. *Nature*, 424(6950), pp.760–763.
- Livermore, P.W., Fournier, A. & Gallet, Y., 2014. Core-flow constraints on extreme archeomagnetic intensity changes. *Earth and Planetary Science Letters*, 387, pp.145–156.
- Olson, P. & Amit, H., 2006. Changes in earth’s dipole. *Naturwissenschaften*, 93(11), pp.519–542. Available at: <http://link.springer.com/10.1007/s00114-006-0138-6>.
- Olson, P. & Christensen, U.R., 2002. The time-averaged magnetic field in numerical dynamos with non-uniform boundary heat flow. *Geophys. J. Int.*, 151, pp.809–823.
- Olson, P. & Deguen, R., 2012. Eccentricity of the geomagnetic dipole caused by lopsided inner core growth. *Nat. Geosci.*, 5, pp.565–569.
- Shaar, R. et al., 2013. Absolute Geomagnetic Field Intensity in Georgia During the Past 6 Millennia. , 3, pp.1–4.
- Shaar, R. et al., 2011. Geomagnetic field intensity: How high can it get? How fast can it change? Constraints from Iron Age copper slag. *Earth and Planetary Science Letters*, 301, pp.297–306.

Shaar, R. et al., 2016. Large geomagnetic field anomalies revealed in Bronze to Iron Age archeomagnetic data from Tel Megiddo and Tel Hazor , Israel. *Earth and Planetary Science Letters*, 442, pp.173–185. Available at: <http://dx.doi.org/10.1016/j.epsl.2016.02.038>.

Tauxe, L. & Yamazaki, T., 2015. *Paleointensities*, Available at: <http://linkinghub.elsevier.com/retrieve/pii/B978044453802400107X>.

Reviewers' Comments:

Reviewer #1 (Remarks to the Author)

The manuscript has improved compared to its previous version. Also, new data published in the meantime (Shaar et al. 2016) have strengthened the case for very high intensities in the Levantine region around 1000 BC and give some indication for a strong inclination anomaly, which answers one of my earlier points. The Levantine intensity high is a very intriguing and enigmatic feature in the Earth's archeomagnetic field. Enigmatic, because it seems to be localized, which is incompatible with any plausible magnetic field structure at the core-mantle boundary. A conceivable explanation given in the paper is that the extreme intensity high was also short-lived (which is very interesting in itself) and that intensity data from locations at moderate angular distance from the Levantine region are simply not sufficiently coeval. The paper explores and discussed implications of, and problems with, the Levantine intensity spike and is worth to be published. The manuscript could be improved by additional explanations in some places and by being more concise in others.

Detailed comments keyed to line numbers

41-42: The sentence is misleading. As shown in the paper, high field intensity cannot be highly spatially localized if its source is at the CMB (and no other source is plausible).

114: Perhaps the choice of dF/dt could be discussed a bit more here. Later in the paper, slightly higher values are used with the argument that the spike is associated with rapid field changes. Could this not justify significantly larger values of dF/dt (say, 0.5 $\mu\text{T}/\text{yr}$) and what would be the consequence?

183: Define x . Only after reading the following sentences several times over, I understood what x is.

191 and following: I do not quite understand why 200 models are needed to show something that seems fairly obvious. One would think that a delta-function spike at the CMB produces the surface anomaly with the smallest possible extent. The large extent associated with a near-delta-spike is already a big problem in light of the available data (Fig. 4d) and it seems obvious that assuming any larger width for the CMB spike can make it only worse. Figs. 5 and 6 do not seem very essential to me. I suggest to shorten the paragraph, keeping mainly the qualitative reasoning.

199-200: While in absolute terms the contribution of the spike is largest to the dipole (at the surface), it is not so in relative terms because the dipole is already very dominant in the background (normal) field.

209-212: I do not understand why lateral variations are brought into play here. In lines 165-167 it is written that

the dynamics is not considered, just the consequence of the existence of a spike. This makes sense and the authors better stick to it at this point of the text.

216-217: Combinations of spikes, including cases with spikes of different polarity, are considered in the supplementary material and do not produce a narrower surface anomaly. Yet, in principle, it should be possible to come up with a CMB magnetic flux distribution that creates a narrow high-intensity field structure at the Earth's surface. Simply start out with prescribing the field of such a structure at the surface, expand it in spherical harmonics up to a degree that it required to give a satisfactory match, and downward continue it to the CMB assuming a source-free mantle. The result at the CMB should be a spike surrounded with concentric rings of alternating negative and positive flux, probably with huge amplitudes. Of course, this is not a likely structure for a dynamo-generated field and might have problems with ohmic dissipation constraints. It may not be worth to point this out in the main text, but it could be mentioned in the supplement.

218-219: It is not obvious that moving the CMB spike away from the location of the highest intensities can help to mitigate in any way the problem with the predicted lateral extent of high intensities, e.g. Fig. 4d.

237: At this point it is not clear that the requirement for a suitable fit to the peak intensity data is used just as a "soft criterion" later and is not formally included in the L1-norm. Could that be done in a sensible way, e.g., by giving these data a strong extra weight?

242: The value $L1 \sim 2$ is at odds with what is shown in Fig.7.

265-267: Can it be understood why moving the spike to the south-east of Israel improves the fit? Is it primarily because this slightly reduces the model intensities in south-east Europe, which are then somewhat less overpredicted? Fig. 4c may suggest this explanation.

298-300: Looking at supplementary figure 6, the predicted inclinations are not much different for Turkey and for Israel. The figure is for a two-spike model, but presumably inclinations in Turkey and Israel are not affected very much by the Texan spike.

Moving part c/d of this figure to the main text (to replace Fig. 5 and/or 6) and adding the few available inclination/declination data in a similar way as intensity data are shown in Fig. 4c, would add to the attractiveness of the paper. If only to show what is to be expected from

possible future directional data if a strong flux spike in the Levantine region exists.

394-409: Means of transport and decay of a spike are discussed, but it may also be of interest to discuss/speculate about ways how a spike can be formed in the first place. For example, one conceivable way could be the expulsion of a very strong toroidal flux concentration (although, in analogy to sunspots, one might here expect a bipolar pair of spikes). Another way could be the concentration of magnetic flux by convergent flow near the

CMB and its intensification by field-line stretching in the associated downwelling.

Reviewer #2 (Remarks to the Author)

The authors made a big effort in their revision. A lot of material (Figs. and text) was added, a lot of corrections were made. Overall they addressed all the issues raised by my first review. I have a few minor comments on their revised manuscript. I therefore recommend publication after minor revisions.

Comments:

1. Lines 19-20: Change "northward and westward" to "westward and then northward" to conform to your text in lines 385-386.
2. Line 22: Add "often neglected in core flow inversions from geomagnetic secular variation" because diffusion is not neglected e.g. in numerical dynamos.
3. Line 30: The sentence as it is makes little sense - obviously the features in the historical field don't reflect extreme events like the spikes because such spikes don't appear in the historical field... Change "extreme" to "long-term" and maybe add a phrase about the possibility that over a longer period such extreme events may arise.
4. Fig. 4a: I suggest to use the same colorbar as in b and c to better illustrate the spike in c.
5. Fig. 4b vs. c: Indeed the spike is localized and the global archeomagnetic field model in Fig. 4b also fails to explain several data points, so no need to blame the spike on all misfits. Note however that the main difference between Figs. 4b and c (apart from the Levant of course) is that in 4b there is an equatorial belt of low intensity values whereas in 4c the belt is broken in southern Africa. The two blue points that fall on this area in west Africa and India are better recovered in 4b...
6. Lines 199-200: I must be misunderstanding something here, the dipole powers at the surface with and without the spike are nearly identical (Fig. 6a). Please clarify.
7. Fig. 6b: The choice of dashed line in a log-scale creates a strange impression of missing symbols in the low degrees. Change to dotted?
8. Line 267: Refer to your Fig. 2c. Note that in Fig. 2 there are stronger CMB patches than the Saudi Arabian that don't produce spikes. Is a low-latitude intense patch a condition for a spike?
9. Line 384: Strange to describe the flow of Finlay et al. (2016) as a gyre of a certain direction in the northern hemisphere, while their gyre is equatorially symmetric. Rewrite simply "meridional advection by a large-scale anti-cyclonic gyre".
10. Lines 387-388: Not clear whether this patch has indeed moved westward, or alternatively has dissipated while the mid-Atlantic lobe has intensified...It is challenging to track patches with 50 years time steps (Amit et al., 2011; Terra-Nova et al., 2015), tracking with time steps of several centuries is really not robust.

Language, typos and very minor comments:

- Line 13: Delete "visible".
- Line 111: Add "uncertainty is".
- Line 114: Change "evidence" to "are evidence for".
- Line 246: Insert space between value to units.
- Line 282: Add "the very thin".

References:

- Amit, H., Korte, M., Aubert, J., Constable, C., Hulot, G., 2011. The time-dependence of intense archeomagnetic flux patches. *J. Geophys. Res.*, 116, B12106, doi:10.1029/2011JB008538.
- Terra-Nova, F., Amit, H., Hartmann, G. A., Trinidade, R. I. F., 2015. The time dependence of reversed archeomagnetic flux patches. *J. Geophys. Res.*, 120, 691-704.

Hagay

Reviewer #3 (Remarks to the Author)

The authors made a substantial and appreciated effort of taking my comments into account. The revised version of this manuscript is now in better shape than the original one. In particular, the authors made a brave attempt at quantifying the uncertainties affecting each datum entering their analysis. They have to work under uncertainty, since the least one can say is that not all the data available around 1000 BC support the spike hypothesis.

Some extra work is needed for the paper to reach out to a potentially broader audience and be considered for publication.

First of all, after reading the revised manuscript, it appears that the abstract is slightly misleading: it seems that the authors are about to provide the coherent link that is missing between the intensity spike and the processes occurring inside the core, an impression that is reinforced by the proposal made at the end of the abstract that the spike grew in place before moving eastward and northward contributing to the growth of the dipole seen in Holocene field models around 1000 BC. There is nothing in the paper to substantiate this statement, except the observation that the current large scale flow inside the core possesses the adequate geometry for this to occur. With a westward drift of 10 km / yr or so, the probability of the large scale flow around 1000 BC to resemble the present-day flow is limited (the large scale eccentric gyre is advected by the drift).

Also, nothing is said about the compatibility of this dynamical scenario with the occurrence of two spikes separated by 200 years, as favoured by R. Shaar and colleagues in their latest study (EPSL, 2016). In passing, in their response to referee 1, the authors argue that this study demonstrates the reproducibility and high fidelity of spikes. But looking in detail, the 2 spikes of the 2016 paper are dated at 980 BC and around 800 BC, the latter being new; a previous spike which supposedly occurred in 890 BC (2011 paper by Shaar and al.) no longer exists, on the account of not meeting selection criteria recently defined by R. Shaar and colleagues. One may say that the 890 BC spike had a brief (spike-like) existence.

Which brings me to the data set: nothing is said in the abstract on the data themselves, and the fact that a great deal of them do not seem to confirm the spike. There is an interesting discussion about this in the paper, but nothing in the abstract points to the potential shortcomings of the dataset and, more importantly, the observational footing on which spikes rest.

In my view, a rewriting of the abstract is in order, for it to convey more precisely the work that is presented here and the conclusions (not speculations) that can be drawn.

Some comments:

1) Regarding the implications of spike geometry for its temporal evolution:

l 398 - l 399: the authors suggest that in Ref 21 Livermore et al. constructed a spike model based on the assumption that the spectrum of the (background + spike) field decreased with increasing degree. This statement is wrong: Livermore et al. considered an ensemble of realizations of background geomagnetic fields on top of which they sought an ensemble of optimized core flows that would generate the largest possible change of geomagnetic intensity at a location at the surface of the Earth. They did not build a (background + spike) field model. Inspection of the spectrum of the SV that their model generates at the surface of the core (their figure 7c) shows on the contrary that the spectrum of the (background + spike) field they could have built based on this SV would have shown increasing power with increasing degree in the range $2 < \ell < 150$. In summary, the citation to Livermore et al. is inappropriate in the context of this sentence.

2) Regarding the last sentence:

'The role of geomagnetic diffusion in controlling decay of geomagnetic spikes can in principle be

tested in geodynamo simulations , although it remains to be seen whether the current generation of models produce features as extreme as the Levanting spike '

(i) what about the role of diffusion in controlling the growth of spikes?

(ii) define the properties which 'extreme' refers too: are you referring to the geometrical properties of your favorite spike models? Or to the timing of the event, which your study did not cover? Or both? Please clarify.

3) What is next? What is your take on spikes?

It is important for the paper to reach out a broad audience that you define a roadmap for geodynamo modellers and archeomagnetists alike. If I am an archeomagnetist, what should I put on top of my to-do list to get that story sorted out? If I am a modeller, what conditions should my numerical model meet in order to be spike-like?

More generally, are there other observations that may help corroborate the spike hypothesis?

4) minor points:

l 196: the component of the field of degree ℓ decays with increasing...

l 309: please use mks units instead of cgs

~

Response to reviewer comments:

We thank the reviewers for their constructive comments, which are addressed in the response below (responses in black). Line numbers in the responses refer to the revised manuscript.

Reviewer #1 (Remarks to the Author):

The manuscript has improved compared to its previous version. Also, new data published in the meantime (Shaar et al. 2016) have strengthened the case for very high intensities in the Levantine region around 1000 BC and give some indication for a strong inclination anomaly, which answers one of my earlier points. The Levantine intensity high is a very intriguing and enigmatic feature in the Earth's archeomagnetic field. Enigmatic, because it seems to be localized, which is incompatible with any plausible magnetic field structure at the core-mantle boundary. A conceivable explanation given in the paper is that the extreme intensity high was also short-lived (which is very interesting in itself) and that intensity data from locations at moderate angular distance from the Levantine region are simply not sufficiently coeval. The paper explores and discussed implications of, and problems with, the Levantine intensity spike and is worth to be published. The manuscript could be improved by additional explanations in some places and by being more concise in others.

Detailed comments keyed to line numbers

41-42: The sentence is misleading. As shown in the paper, high field intensity cannot be highly spatially localized if its source is at the CMB (and no other source is plausible).

Response: We have rephrased the sentence accordingly (new line number 46).

114: Perhaps the choice of dF/dt could be discussed a bit more here. Later in the paper, slightly higher values are used with the argument that the spike is associated with rapid field changes. Could this not justify significantly larger values of dF/dt (say, $0.5 \mu\text{T/yr}$) and what would be the consequence?

Response: We investigated this issue using the value of $\partial F/\partial t = 0.62 \mu\text{T/yr}$ suggested by Livermore et al (ref. 22 of the main text). The result is shown in Figure R1 below, which shows that increasing $\partial F/\partial t$ simply shifts the data distribution shown in the new Figure 6 (old Figure 7) of the main text to lower L^1 . This arises because increasing $\partial F/\partial t$ gives more weight to the age uncertainties, which already represent the main contribution to σ_t for $\partial F/\partial t = 0.15 \mu\text{T/yr}$ (Figure 3b of the main text). We now note this fact on line 245 of the revised manuscript.

Figure R1. Same as the new Figure 6 of the main text but for $\partial F/\partial t = 0.62 \mu\text{T/yr}$.

183: Define x. Only after reading the following sentences several times over, I understood what x is.

Response: We now state on 187 that x is a parameter that is > 1 .

191 and following: I do not quite understand why 200 models are needed to show something that seems fairly obvious. One would think that a delta-function spike at the CMB produces the surface anomaly with the smallest possible extent. The large extent associated with a near-delta-spike is already a big problem in light of the available data (Fig. 4d) and it seems obvious that assuming any larger width for the CMB spike can make it only worse. Figs. 5 and 6 do not seem very essential to me. I suggest to shorten the paragraph, keeping mainly the qualitative reasoning.

Response: It is true that 200 models are not required to obtain the result in Figure 5; this simply represents the number of models run in the course of our tests. We agree with the referee that Figure 5 is not essential and have removed it along with the accompanying text;

however, the old Figure 6 was requested by referees 2 and 3 in the previous revision and so we wish to keep it. We now refer to Supplementary Figure 4, which shows an example model with $A = 500 \mu\text{T}$ and $\sigma = 1^\circ$, when discussing the minimum spike width (line 194) as this does not include a background field; by comparison, the background field included in the calculation shown in Figure 4c of the main text significantly increases the predicted spike width.

199-200: While in absolute terms the contribution of the spike is largest to the dipole (at the surface), it is not so in relative terms because the dipole is already very dominant in the background (normal) field.

Response: We agree and have added a comment to this effect on line 201-202.

209-212: I do not understand why lateral variations are brought into play here. In lines 165-167 it is written that the dynamics is not considered, just the consequence of the existence of a spike. This makes sense and the authors better stick to it at this point of the text.

Response: We agree with the referee and have removed the sentence on lateral variations.

216-217: Combinations of spikes, including cases with spikes of different polarity, are considered in the supplementary material and do not produce a narrower surface anomaly. Yet, in principle, it should be possible to come up with a CMB magnetic flux distribution that creates a narrow high-intensity field structure at the Earth's surface. Simply start out with prescribing the field of such a structure at the surface, expand it in spherical harmonics up to a degree that it required to give a satisfactory match, and downward continue it to the CMB assuming a source-free mantle. The result at the CMB should be a spike surrounded with concentric rings of alternating negative and positive flux, probably with huge amplitudes. Of course, this is not a likely structure for a dynamo-generated field and might have problems with ohmic dissipation constraints. It may not be worth to point this out in the main text, but it could be mentioned in the supplement.

Response: We agree with the referees' assessment and have added statements to this effect in the supplementary information.

218-219: It is not obvious that moving the CMB spike away from the location of the highest intensities can help to mitigate in any way the problem with the predicted lateral extent of high intensities, e.g. Fig. 4d.

Response: The new Figure 6 (old Figure 7) shows that moving the spike away from the location of highest intensities does indeed reduce the misfit. The reason for this, suggested by the referee in their comment (lines 265-267) below, has been added to the manuscript on line 267.

237: At this point it is not clear that the requirement for a suitable fit to the peak intensity data is used just as a "soft criterion" later and is not formally included in the L1-norm. Could that be done in a sensible way, e.g., by giving these data a strong extra weight?

Response: To address this point we have done two things. First, we took the 143 data and removed the age uncertainties for the two points that have $F > 100 \mu\text{T}$ (Figure R2). These two data points have $\sigma_{lab} = 10.1$ and $16 \mu\text{T}$, which are some of the largest values in the

database (black points in Figure 3b) and the resulting σ_t values are comparable to the majority of data. Hence this test produced little change in L^1 compared to the case where age uncertainties were included for all data (compare Figure 6 and Figure R2).

Figure R2. Same as the new Figure 6 of the main text but with age uncertainties removed on the two data (of 143) with $F > 100 \mu\text{T}$.

In the second test we arbitrarily assigned the two data points with $F > 100 \mu\text{T}$ a value of $\sigma_t = 0.01 \mu\text{T}$, which is clearly unrealistically low. The resulting L^1 values (Figure R3) are very high as the model misfit is high for many of the other data, even though it can fit the spike data well. We conclude that the method already employed for finding the best-fitting model is adequate given the relatively sparse dataset and attendant age uncertainties and have kept the method description in the original manuscript.

Figure R3. Same as the new Figure 6 of the main text but with $\sigma_t = 0.01 \mu\text{T}$ for the two data (of 143) with $F > 100 \mu\text{T}$.

242: The value $L^1 \sim 2$ is at odds with what is shown in Fig.7.

Response: The new Figure 6 (old Figure 7) uses $\frac{\partial F}{\partial t} = 0.2 \mu\text{T/yr}$ while the quoted value of $L^1 \sim 2$ is for $\frac{\partial F}{\partial t} = 0.15 \mu\text{T/yr}$.

265-267: Can it be understood why moving the spike to the south-east of Israel improves the fit? Is it primarily because this slightly reduces the model intensities in south-east Europe, which are then somewhat less overpredicted? Fig. 4c may suggest this explanation.

Response: We agree with the referees' assessment and have included this explanation on line 267.

298-300: Looking at supplementary figure 6, the predicted inclinations are not much different for Turkey and for Israel. The figure is for a two-spike model, but presumably inclinations in Turkey and Israel are not affected very much by the Texan spike. Moving part c/d of this

figure to the main text (to replace Fig. 5 and/or 6) and adding the few available inclination/declination data in a similar way as intensity data are shown in Fig. 4c, would add to the attractiveness of the paper. If only to show what is to be expected from possible future directional data if a strong flux spike in the Levantine region exists.

Response: We have merged the discussion of the dual spike model in the supplement with that in the main text (paragraph starting on line 338) along with a modified version of Supplementary Figure 6 (new Figure 8), which now includes the inclination and declination data as suggested by the referee.

394-409: Means of transport and decay of a spike are discussed, but it may also be of interest to discuss/speculate about ways how a spike can be formed in the first place. For example, one conceivable way could be the expulsion of a very strong toroidal flux concentration (although, in analogy to sunspots, one might here expect a bipolar pair of spikes). Another way could be the concentration of magnetic flux by convergent flow near the CMB and its intensification by field-line stretching in the associated downwelling.

Response: We thank the referee for these interesting suggestions and agree that formation of the spike is an important issue. We have added a brief discussion of these potential formation mechanisms to the final paragraph of the paper where we consider future directions.

Reviewer #2 (Remarks to the Author):

The authors made a big effort in their revision. A lot of material (Figs. and text) was added, a lot of corrections were made. Overall they addressed all the issues raised by my first review. I have a few minor comments on their revised manuscript. I therefore recommend publication after minor revisions.

Comments:

1. Lines 19-20: Change “northward and westward” to “westward and then northward” to conform to your text in lines 385-386.

Response: We have made the required change on line 23.

2. Line 22: Add “often neglected in core flow inversions from geomagnetic secular variation” because diffusion is not neglected e.g. in numerical dynamos.

Response: We have made the required change on line 25.

3. Line 30: The sentence as it is makes little sense - obviously the features in the historical field don't reflect extreme events like the spikes because such spikes don't appear in the historical field... Change “extreme” to “long-term” and maybe add a phrase about the possibility that over a longer period such extreme events may arise.

Response: We have changes “extreme” to “long-term” as suggested (line 23).

4. Fig. 4a: I suggest to use the same colorbar as in b and c to better illustrate the spike in c.

Response: We have made the required change.

5. Fig. 4b vs. c: Indeed the spike is localized and the global archeomagnetic field model in Fig. 4b also fails to explain several data points, so no need to blame the spike on all misfits. Note however that the main difference between Figs. 4b and c (apart from the Levant of course) is that in 4b there is an equatorial belt of low intensity values whereas in 4c the belt is broken in southern Africa. The two blue points that fall on this area in west Africa and India are better recovered in 4b...

Response: The blue points referred to here, Mali and India, are both nominally at younger ages than the Israeli and Jordanian data and so we might not expect to fit them as well with the spike. Also, the difference in F in these locations with and without the spike field is much smaller ($\sim 15\mu T$) than the difference in the Levant ($\sim 50\mu T$), so this is a small effect by comparison. We state on line 253 that the model cannot simultaneously match the high and low data in the Levantine region and note on line 260 that the data from Mali and India are younger than the spike.

6. Lines 199-120: I must be misunderstanding something here, the dipole powers at the surface with and without the spike are nearly identical (Fig. 6a). Please clarify.

Response: The power in the dipole component of the spike field is approximately 50 times less than the power in the background dipole field shown in Figure 5. The power is a quadratic measure of field strength and so the actual spike field is approximately 15% of the background field. This is now noted in the text on line 389.

7. Fig. 6b: The choice of dashed line in a log-scale creates a strange impression of missing symbols in the low degrees. Change to dotted?

Response: We have reduced the weight of the red line so that this effect does not arise, while still allowing it to be seen alongside the black line.

8. Line 267: Refer to your Fig. 2c. Note that in Fig. 2 there are stronger CMB patches than the Saudi Arabian that don't produce spikes. Is a low-latitude intense patch a condition for a spike?

Response: We have added the reference to Fig 2c to line 266. The remaining issue relates to the definition of a spike. We have defined the Levantine spike in terms of its observed intensity variation at Earth's surface - a factor 2 rise and fall over an area of only $\approx 20^\circ$ longitude - and this does not require that the spike be located at low latitude. Since the dipole field strength increases with latitude, spikes at higher latitudes would be associated with very high intensities by our definition. Alternatively, one could define spikes by their generation mechanism and this may distinguish between high- and low-latitude features. However, since our model does not consider the dynamics of spikes we do not pursue this further. We have added a discussion of future directions to the final paragraph of the paper.

9. Line 384: Strange to describe the flow of Finlay et al. (2016) as a gyre of a certain direction in the northern hemisphere, while their gyre is equatorially symmetric. Rewrite simply "meridional advection by a large-scale anti-cyclonic gyre".

Response: We have made the required change on line 391.

10. Lines 387-388: Not clear whether this patch has indeed moved westward, or alternatively has dissipated while the mid-Atlantic lobe has intensified...It is challenging to track patches with 50 years time steps (Amit et al., 2011; Terra-Nova et al., 2015), tracking with time steps of several centuries is really not robust.

Response: We have added a statement to line 397 acknowledging the difficulty described by the referee and included the suggested citations.

Language, typos and very minor comments:

- Line 13: Delete “visible”.

Response: We have made the required change.

- Line 111: Add “uncertainty is”.

Response: We have made the required change.

- Line 114: Change “evidence” to “are evidence for”.

Response: We have made the required change.

- Line 246: Insert space between value to units.

Response: We have made the required change.

- Line 282: Add “the very thin”.

Response: We have made the required change.

References:

- Amit, H., Korte, M., Aubert, J., Constable, C., Hulot, G., 2011. The time-dependence of intense archeomagnetic flux patches. *J. Geophys. Res.*, 116, B12106, doi:10.1029/2011JB008538.

- Terra-Nova, F., Amit, H., Hartmann, G. A., Trindade, R. I. F., 2015. The time dependence of reversed archeomagnetic flux patches. *J. Geophys. Res.*, 120, 691-704.

Hagay

Reviewer #3 (Remarks to the Author):

The authors made a substantial and appreciated effort of taking my comments into account. The revised version of this manuscript is now in better shape than the original one. In particular, the authors made a brave attempt at quantifying the uncertainties affecting each datum entering their analysis. They have to work under uncertainty, since the least one can say is that not all the data available around 1000 BC support the spike hypothesis.

Some extra work is needed for the paper to reach out to a potentially broader audience and be considered for publication.

First of all, after reading the revised manuscript, it appears that the abstract is slightly misleading: it seems that the authors are about to provide the coherent link that is missing between the intensity spike and the processes occurring inside the core, an impression that is reinforced by the proposal made at the end of the abstract that the spike grew in place before moving eastward and northward contributing to the growth of the dipole seen in Holocene field models around 1000 BC. There is nothing in the paper to substantiate this statement, except the observation that the current large scale flow inside the core possesses the adequate geometry for this to occur. With a westward drift of 10 km / yr or so, the probability of the large scale flow around 1000 BC to resemble the present-day flow is limited (the large scale eccentric gyre is advected by the drift).

Response: We have changed the statement containing the phrase ‘coherent link’ so that we do not refer to processes in the core. Regarding the last point, it is not necessarily the case that the gyre is advected westward. Indeed, a steady retrograde differential rotation would continually advect magnetic field lines westward assuming that diffusion is negligible. The geomagnetic secular variation is more complex than a simple westward drift and the core flow imagined from it contains information on a variety of dynamical processes, including bulk fluid motion and waves, but much of the signal (clearly excluding rapid variations) can be explained by a steady flow pattern. It therefore seems perfectly reasonable to suggest that a flow pattern similar to the present-day gyre operated around the time of the spike.

Also, nothing is said about the compatibility of this dynamical scenario with the occurrence of two spikes separated by 200 years, as favoured by R. Shaar and colleagues in their latest study (EPSL, 2016).

Response: We have added a paragraph to the discussion starting on line 403.

In passing, in their response to referee 1, the authors argue that this study demonstrates the reproducibility and high fidelity of spikes. But looking in detail, the 2 spikes of the 2016 paper are dated at 980 BC and around 800 BC, the latter being new; a previous spike which supposedly occurred in 890 BC (2011 paper by Shaar and al.) no longer exists, on the account of not meeting selection criteria recently defined by R. Shaar and colleagues. One may say that the 890 BC spike had a brief (spike-like) existence.

Response: We now note this issue in the discussion in the new paragraph starting on line 403.

Which brings me to the data set: nothing is said in the abstract on the data themselves, and the fact that a great deal of them do not seem to confirm the spike. There is an interesting discussion about this in the paper, but nothing in the abstract points to the potential shortcomings of the dataset and, more importantly, the observational footing on which spikes rest.

In my view, a rewriting of the abstract is in order, for it to convey more precisely the work that is presented here and the conclusions (not speculations) that can be drawn.

Response: We agree with the issues raised by the referee and have rewritten the abstract to specifically discuss the data as well as the other items detailed above.

Some comments:

1) Regarding the implications of spike geometry for its temporal evolution:

1 398 - 1 399: the authors suggest that in Ref 21 Livermore et al. constructed a spike model based on the assumption that the spectrum of the (background + spike) field decreased with increasing degree. This statement is wrong: Livermore et al. considered an ensemble of realizations of background geomagnetic fields on top of which they sought an ensemble of optimized core flows that would generate the largest possible change of geomagnetic intensity at a location at the surface of the Earth. They did not build a (background + spike) field model. Inspection of the spectrum of the SV that their model generates at the surface of the core (their figure 7c) shows on the contrary that the spectrum of the (background + spike)

field they could have built based on this SV would have shown increasing power with increasing degree in the range $2 < \ell < 150$. In summary, the citation to Livermore et al. is inappropriate in the context of this sentence.

Response: We thank the referee for correcting this error and have removed the citation to Livermore et al.

2) Regarding the last sentence:

'The role of geomagnetic diffusion in controlling decay of geomagnetic spikes can in principle be tested in geodynamo simulations, although it remains to be seen whether the current generation of models produce features as extreme as the Levanting spike'

(i) what about the role of diffusion in controlling the growth of spikes?

(ii) define the properties which 'extreme' refers to: are you referring to the geometrical properties of your favorite spike models? Or to the timing of the event, which your study did not cover? Or both? Please clarify.

Response: In the paragraph starting on line 4121 we speculate, based on the Ohmic dissipation associated with our synthetic model spikes, that radial diffusion could be important in determining the evolution of the spike. Radial diffusion cannot be constrained by geomagnetic observations, but can be (and has been) investigated in numerical dynamo models and here we suggest this as a direction for future work, assuming that spikes are actually present in the dynamo models. By spikes we are thinking of spatial features like that in Figure 4c of the paper and we have reworded the sentence to make this clear.

3) What is next? What is your take on spikes?

It is important for the paper to reach out a broad audience that you define a roadmap for geodynamo modellers and archeomagnetists alike. If I am an archeomagnetist, what should I put on top of my to-do list to get that story sorted out? If I am a modeller, what conditions should my numerical model meet in order to be spike-like?

More generally, are there other observations that may help corroborate the spike hypothesis?

Response: The final paragraph of the manuscript focuses on future directions and so we have extended this to cover the issues raised by the referee.

4) minor points:

l 196: the component of the field of degree ℓ decays with increasing...

l 309: please use mks units instead of cgs

Reviewers' Comments:

Reviewer #1:

Remarks to the Author:

The authors responded in a satisfactory way to the points I raised. I have not further comments. This is an interesting paper, which in my view is now ready for publication.

Reviewer #2:

Remarks to the Author:

The authors have addressed well all the comments from my previous review. I have no further comments. I recommend publication of the manuscript as it is.

Best regards,

Hagay

Reviewer #3:

Remarks to the Author:

I thank the authors for addressing the issues I raised over the various rounds of review. Their study is now worth publishing. I hope that it will stimulate further work on those enigmatic geomagnetic spikes.